# Learning Mamba as a Continual Learner

## Abstract

Continual learning (CL) aims to efficiently learn and accumulate knowledge from a data stream with different distributions. By formulating CL as a sequence prediction task, meta-continual learning (MCL) enables to meta-learn an efficient continual learner based on the recent advanced sequence models, *e.g.*, Transformers. Although attention-free models (*e.g.*, Linear Transformers) can ideally match CL's essential objective and efficiency requirements, they usually perform not well in MCL. Considering that the attention-free Mamba achieves excellent performances matching Transformers' on general sequence modeling tasks, in this paper, we aim to answer a question – *Can attention-free Mamba perform well on MCL?* By formulating Mamba with a selective state space model (SSM) for MCL tasks, we propose to meta-learn Mamba as a continual learner, referred to as **MambaCL**. By incorporating a selectivity regularization, we can effectively train MambaCL. Through comprehensive experiments across various CL tasks, we also explore *how Mamba and other models perform in different MCL scenarios.* Our experiments and analyses highlight the promising performance and generalization capabilities of Mamba in MCL.

## 1 Introduction

Continual learning (CL) aims to efficiently learn and accumulate knowledge in a non-stationary data stream (De Lange et al., 2021; Wang et al., 2024) containing different tasks. Given a sequence of data $\mathcal{D}_T = ((\mathbf{x}_1, y_1), ..., (\mathbf{x}_t, y_t), ..., (\mathbf{x}_T, y_T))$ with a series of paired observations $\mathbf{x}_i$ (*e.g.*, images) and targets $y_i$ (*e.g.*, class labels) from different tasks, CL is usually produced to learn one model $P_{\phi_t}(y|\mathbf{x})$ parameterized by $\phi_t$ that can perform prediction for any tasks corresponding to the seen data $\mathcal{D}_t$. For example, in class incremental learning (CIL) (Rebuffi et al., 2017; Zhou et al., 2023), a widely studied CL scenario, $\mathcal{D}_T$ consists of data with incrementally added classes, and $P_{\phi_t}(y|\mathbf{x})$ is trained to recognize all previously seen classes. To ensure computational and memory efficiency, CL methods are explored for learning from data streams while minimizing the storage of historical data or limiting running memory growth, such as restricting the increase rate to be constant or sub-linear (De Lange et al., 2021; Ostapenko et al., 2021). The main challenge in CL is to preserve performance on previously seen tasks while continually updating the model parameters $\phi_t$ (De Lange et al., 2021; Wang et al., 2024).

CL methods continually train/update the model $P_{\phi_t}(y|\mathbf{x})$ from seen sequence $\mathcal{D}_t$ at arbitrary step $t$ and perform predictions on any observation $\mathbf{x}^{\text{test}}$ (following seen data distribution) for the corresponding $y^{\text{test}}$. From this perspective, the whole learning and inference process in CL can be seen as a sequence prediction (SP) problem, *i.e.*, predicting $y_{\text{test}}$ of a query $\mathbf{x}_{\text{test}}$ conditioning on the seen data sequence and the testing input, *i.e.*, $(\mathcal{D}_t^{\text{train}}, \mathbf{x}^{\text{test}}) \equiv (\mathbf{x}_1^{\text{train}}, y_1^{\text{train}}, ..., \mathbf{x}_t^{\text{train}}, y_t^{\text{train}}, \mathbf{x}^{\text{test}})$ (Lee et al., 2024; Bornschein et al., 2024). In conventional CL, the model parameter $\phi_t$ is trained to maintain the states on the sequence, *i.e.*, knowledge in the historical data, in a way of $\phi_{t+1} = \texttt{optim-step}(\phi_t, \mathbf{x}_t, y_t)$. This connection between sequence prediction and CL training process motivates us to investigate meta-learning a *continual learner* as a sequence prediction model, for computation-and-data-efficient CL. Through meta-continual learning (MCL) framework (Lee et al., 2024; Son et al., 2023), a continual learner $f_\theta()$ parameterized by $\theta$ is trained via sequence prediction on multiple CL episodes. A meta-learned $f_\theta()$ can take a given sequence $(\mathcal{D}_t, \mathbf{x}_{t+1})$ as input and predict the label $y_{t+1} = f_\theta((\mathcal{D}_t, \mathbf{x}_{t+1}))$, which is equivalent to a predictive model conditioning on the seen data stream $P_\theta(y|\mathbf{x}, \mathcal{D}_t)$. The data stream can also be seen as a *context* of the tasks for performing prediction for a new query.

Transformers (Vaswani et al., 2017; Touvron et al., 2023) have shown strong sequence modeling capabilities and next-token prediction performance in language modeling (LM), relying on self-attention across per-step tokens and emergent in-context learning (ICL) ability (Brown et al., 2020; Garg et al., 2022). It is thus straightforward to meta-learn a Transformer as the SP-based continual learner (Lee et al., 2024; Son et al., 2023). Given a data stream in CL, a meta-learned Transformer generates a new key-value pair at each step and makes the prediction for each query based on attention over the key-value pairs retained from all preceding training samples. Benefiting from the retrieval-based modeling, Transformers can perform effectively in continual learning (CL) (Lee et al., 2024; Bornschein et al., 2024). However, they require maintaining key-value pairs for all seen training samples in a *key-value cache*, allowing the model to access *all* seen samples during inference. It contradicts the principles and intended purpose of continual learning. Although the key-value cache can be viewed as the hidden state of a recurrent neural network (RNN) (Katharopoulos et al., 2020; Lee et al., 2024), analogous to the parameters of a learner, its size grows linearly with the number of all seen tokens and suffers from increasing memory and computational demands over time. Despite their advanced sequence modeling capabilities as in (Lee et al., 2024), Transformers may not be an ideal choice for continual learning due to misalignment with the objectives of CL and efficiency concerns. A series of attention-free models achieve efficiency by approximating the softmax attention with kernel methods and linear operations, leading to *constant* hidden state sizes and *linear* computation complexity, such as Linear Transformer (Katharopoulos et al., 2020) and Performer (Choromanski et al., 2020). Although these efficient Transformers align better with the purpose of CL, it is seen that they cannot perform well in MCL (Lee et al., 2024), due to limitations in approximation and insufficient expressive power (Katharopoulos et al., 2020; Choromanski et al., 2020; Tay et al., 2020).

Recent advancements of the state space models (SSMs) on sequence modeling lead to a series of attention-free models that are efficient in processing long sequences with nearly linear computation (Gu et al., 2021a;b). By integrating time-varying modeling into the SSM as a selective SSM, Mamba (Gu & Dao, 2023; Dao & Gu, 2024) can achieve near state-of-the-art performances on sequence modeling tasks (*e.g.*, LM tasks (Gao et al., 2020)). Given its exceptional performance as an attention-free model with a constant hidden state size, which ideally aligns with the requirements of MCL, rather than relying on Transformers Lee et al. (2024), we pose a concrete question: *Can the attention-free model Mamba perform well in MCL?* In this paper, we investigate this question by formulating the selective SSM and Mamba to handle MCL, referred to as **MambaCL**. We identify that it is not trivial to train the sequence prediction models, including Mamba, for MCL, due to difficulty in convergence. To address the issue, we introduce a selectivity regularizer relying on the connection across SSM/Mamba and Linear Transformers and Transformers, which guides the behaviour of the generated time-variant parameters of the selective SSM during training. Relying on the specifically designed regularization and customized designs, we achieve an effective MambaCL model for MCL. Beyond the scope of the existing work (Lee et al., 2024) focusing on basic MCL formulation and setting, we expand the formulation and studies to more realistic scenarios and try to answer – *how can different models (including Transformers and Mamba) perform in different MCL tasks.* Our experiments and analyses show that Mamba can perform well on most of the MCL scenarios. Mamba performs significantly better than other attention-free methods, *e.g.*, Linear Transformers; Mamba can match or outperform the performances of Transformers with fewer parameters and computations. Specifically, on some challenging with more global structures across the sequences (*e.g.*, fine-grained data) and many challenging scenarios (*e.g.*, domain shifts and long sequences), Mamba can perform more reliably and effectively than Transformers, demonstrating better generalization and robustness. Additionally, we analyzed the influence of the model design and conducted preliminary studies to explore the potential of model variants of Mamba, *e.g.*, Mamba mixture-of-experts (MoE), in MCL.

## 2 RELATED WORK

**Continual learning** focuses on mitigating catastrophic forgetting, a significant challenge in model training across sequential tasks (De Lange et al., 2021; Wang et al., 2024). The predominant approaches to continual learning are categorized into three main types: replay-based, regularization-based, and architecture-based methods. Replay-based methods, such as maintaining a memory buffer for old task data, effectively prevent forgetting but are constrained by buffer size and potential privacy issues (Rebuffi et al., 2017; Lopez-Paz & Ranzato, 2017; Chaudhry et al., 2019; Buzzega et al., 2020). Alternatively, generative models can approximate previous data distributions to pro-

duce pseudo-samples (Shin et al., 2017; Rostami et al., 2019; Riemer et al., 2019). Regularization-based strategies (Kirkpatrick et al., 2017; Zenke et al., 2017; Nguyen et al., 2017; Li & Hoiem, 2017; Aljundi et al., 2018; Zhang et al., 2020) mitigate forgetting by penalizing changes to critical parameters of previous tasks and employing knowledge distillation to retain earlier knowledge. Lastly, architecture-based methods (Yoon et al., 2017; Serra et al., 2018; Li et al., 2019; Yan et al., 2021; Ye & Bors, 2023) allocate specific subsets of parameters to individual tasks, utilizing techniques like task masking or dynamic architecture adjustment to minimize task interference.

**Meta-learning** is a learning paradigm where models improve their ability to adapt to new tasks by leveraging limited data and prior experience. The bi-level optimization framework of meta-learning is inherently suited for continual learning, as it focuses on balancing the fit for current tasks while maintaining generalization across all previously encountered tasks (Riemer et al., 2018; Beaulieu et al., 2020; Gupta et al., 2020; Wu et al., 2024). **Meta-continual learning** (MCL) deviates from traditional continual learning settings by incorporating multiple continual learning episodes, structured into meta-training and meta-testing sets (Son et al., 2023). Lee et al. (2024) conceptualizes MCL as a sequence modeling problem, aligning the continual learning objectives with autoregressive models typical in language modeling. OML (Javed & White, 2019) employs a dual-architecture approach, updating a prediction network while keeping the encoder static during training, then optimizing both components in meta-testing for stability. MetaICL (Min et al., 2022) introduces a meta-training framework for natural language in-context learning. MetaICL sharing a common mathematical formulation with MCL, while the underlying functions to be fitted are distinct. Compared to text sequences, the problems we address are inherently more complex, requiring the learning of more intricate functions and making the learning process more challenging.

**Transformer** architecture is esteemed for its superior sequence modeling capabilities, largely attributed to its attention mechanism (Vaswani et al., 2017). Decoder-only models like GPT (Brown et al., 2020) and Llama (Touvron et al., 2023), which process inputs causally, have significantly propelled the success of modern deep learning. Although Transformers employing softmax-based attention benefit from efficient parallel training, they encounter challenges due to their quadratic computational complexity relative to sequence length. This has prompted a shift towards more RNN-like models capable of linear-time sequence modeling. As a viable alternative, linear attention substitutes the traditional exponential similarity function with a simple dot product across transformed key/query vectors, gaining traction through recent advancements (Katharopoulos et al., 2020; Choromanski et al., 2020; Tay et al., 2020).

**State Space Models** (SSMs), inspired by traditional state-space models (Kalman, 1960), have recently emerged as a promising architecture for sequence modeling (Gu et al., 2021a;b). Mamba incorporates time-varying parameters into the SSM framework through a selective architecture and enhances training and inference efficiency with a hardware-aware algorithm (Gu & Dao, 2023; Dao & Gu, 2024). It is widely applied in various domains such as computer vision, natural language processing, and speech, etc.

## 3 PROBLEM FORMULATION AND METHODOLOGY

In **Continual Learning** (CL), given a non-stationary data stream $\mathcal{D}_T^{\text{train}} = ((\mathbf{x}_1, y_1), \ldots, (\mathbf{x}_t, y_t), \ldots, (\mathbf{x}_T, y_T))$ as training data, where $\mathbf{x}_t \in \mathcal{X}_t$ and $y_t \in \mathcal{Y}_t$. A predictive model $g_{\phi_t}() : \mathcal{X} \to \mathcal{Y}$ is trained on the stream (at step $t$) as $P_{\phi_t}(y|\mathbf{x})$ for a potential testing set $\mathcal{D}_T^{\text{test}} = \{(\mathbf{x}_n, y_n)\}_{n=1}^N$, where $\mathbf{x}_t \in \mathcal{X}_t$ and $y_t \in \mathcal{Y}_t$, with the same distribution as the training set. A conventional continual learner is manually crafted for continual updating/optimizing the model parameter $\phi_t$. The data stream $\mathcal{D}_T$ usually consists of the data from different tasks or distributions, which is usually piecewise stationary within an interval of a task. In a general online CL setting, each sample point can only be seen once; if the samples belonging to one task can be held and accessed as a batch, it is an offline CL. We mainly consider the online CL setting.

For achieving efficient CL, **Meta-Continual Learning** (MCL) (Lee et al., 2024; Son et al., 2023) is formulated to meta-learn a parameterized continual learner that can efficiently learn/update a predictive model (in $\mathcal{X} \to \mathcal{Y}$) from samples in a data stream (in $\mathcal{X} \times \mathcal{Y}$). Considering that the (continually) learned model from a sequence $\mathcal{D}_t^{\text{train}}$ is deployed for prediction given a testing sample input $\mathbf{x}_{\text{test}}$, i.e., $P_\theta(y_{\text{test}}|\mathbf{x}_{\text{test}}, \mathcal{D}_t^{\text{train}})$. Thus MCL is equivalent to learning a functional model of the predictive model functions. And MCL can be treated as the task of learning a **sequence prediction** model $f_\theta() : (\mathcal{X} \times \mathcal{Y}) \times \mathcal{X} \to \mathcal{Y}$ parameterized by $\theta$. $f_\theta()$ can continually take streaming data

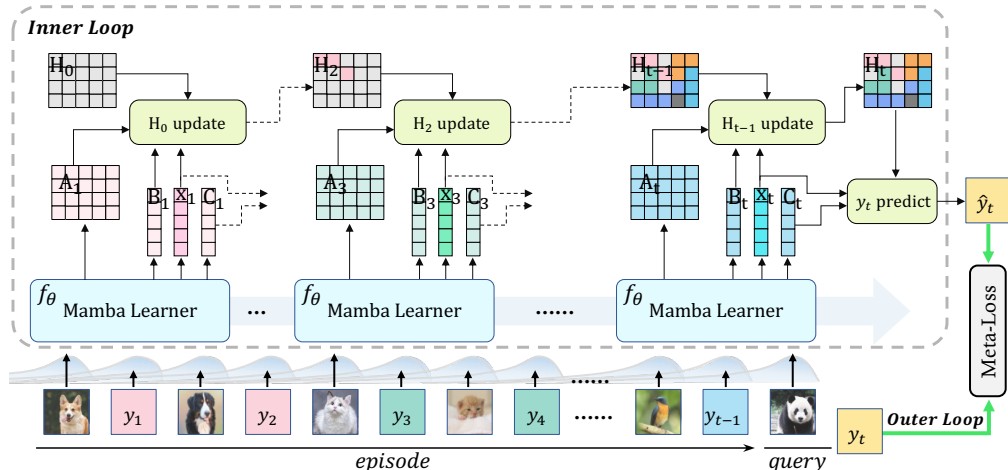

Figure 1: The overall framework of our proposed methods. We meta-train a Mamba Learner $f_\theta()$ to perform meta-continual learning (MCL) by processing an online data stream containing paired $(\mathbf{x}, y)$ examples. Meta-learning of this continual learner is conducted across multiple CL episodes. The model produces predictions by relying on the retained hidden state. Here, we demonstrate how the Mamba learner recurrently processes input data at steps 0, 2, and $t-1$, respectively.

$\mathcal{D}_t^{\text{train}}$ as input and make predictions for any testing samples in a $\mathcal{D}^{\text{test}}$ conditioning on $\mathcal{D}_t^{\text{train}}$ via $\hat{y}_{\text{test}} = f_\theta(\mathbf{x}_{\text{test}}, \mathcal{D}_t^{\text{train}})$. The learner updates internal hidden states to reflect the continually taken data samples corresponding to a CL process. By giving multiple episodes with $(\mathcal{D}^{\text{train}}, \mathcal{D}^{\text{test}})$, the parameter $\theta$ of the learner can be learned in the meta-learning/updating process for optimizing the performance on all $\mathcal{D}^{\text{test}}$. Note that the targets $y$ in different episodes are independent, which are only symbolic indicators without general semantic meaning across episodes. In this work, we focus on MCL based on a parameterized sequence prediction model for general purpose, despite the existence of other types of meta-learning scheme (Finn et al., 2017; Javed & White, 2019).

## 3.1 PRELIMINARIES: TRANSFORMERS, LINEAR TRANSFORMERS, AND SSMS

**Transformers** produce next-token predictions in sequence relying on a self-attention mechanism (Vaswani et al., 2017). Given a sequence of $N$ vectors in $M$-dimension denoted as $\mathbf{Z} \in \mathbb{R}^{N \times M}$, the vanilla self-attention is formulate with a *softmax attention* method:

$$\mathbf{Q} = \mathbf{Z}\mathbf{W}_Q, \ \mathbf{K} = \mathbf{Z}\mathbf{W}_K, \ \mathbf{V} = \mathbf{Z}\mathbf{W}_V, \ \mathbf{u}_t = \sum_{j=1}^{N} \frac{\exp\left(\mathbf{Q}_t \mathbf{K}_j^\top / \sqrt{d}\right)}{\sum_{j=1}^{N} \exp\left(\mathbf{Q}_t \mathbf{K}_j^\top / \sqrt{d}\right)} \mathbf{V}_j, \quad (1)$$

where $\mathbf{W}_Q \in \mathbb{R}^{M \times C}$, $\mathbf{W}_K \in \mathbb{R}^{M \times C}$, $\mathbf{W}_V \in \mathbb{R}^{C \times C}$ are the projection weight matrices, $\mathbf{u}_t \in \mathbb{R}^C$ denote the output embedding, and $C$ is the hidden dimension. $\mathbf{Q}_t$, $\mathbf{K}_j$, and $\mathbf{V}_j$ denote the indexed vectors in the corresponding matrices. The notation fonts are slightly abused to be consistent with the literatures. Each input token generates a key-value pair, leading to a linearly increased key-value cache size. Softmax attention measures the similarities between the query-key pairs, leading to $\mathcal{O}(N^2)$ complexity.

**Linear Transformer** (Katharopoulos et al., 2020) reduces the complexity relying a *linear attention* method. By applying a feature representation function $\phi()$ corresponding to a kernel for $\mathbf{Q}$ and $\mathbf{K}$, the linear attention method replaces the softmax attention with a *linear operation* as:

$$\mathbf{u}_t = \sum_{j=1}^{N} \frac{\mathbf{Q}_t \mathbf{K}_j^\top}{\sum_{j=1}^{N} \mathbf{Q}_t \mathbf{K}_j^\top} \mathbf{V}_j = \frac{\mathbf{Q}_t \left(\sum_{j=1}^{N} \mathbf{K}_j^\top \mathbf{V}_j\right)}{\mathbf{Q}_t \left(\sum_{j=1}^{N} \mathbf{K}_j^\top\right)}, \quad (2)$$

where $\mathbf{Q} = \phi(\mathbf{Z}\mathbf{W}_Q)$, $\mathbf{K} = \phi(\mathbf{Z}\mathbf{W}_K)$, $\mathbf{V} = \mathbf{Z}\mathbf{W}_V$, $\phi(\cdot)$ is set as $\phi(\mathbf{x}) = \text{elu}(\mathbf{x}) + 1$ in (Katharopoulos et al., 2020). **Performer** employs $\phi(\mathbf{x}) = \exp\left(\mathbf{x}\mathbf{W}_p - \|\mathbf{x}\|^2/2\right)$, with $\mathbf{W}_p$ comprising orthogonal random vectors (Choromanski et al., 2020). Through rearranging $(\mathbf{Q}\mathbf{K}^\top)\mathbf{V}$ as $\mathbf{Q}(\mathbf{K}^\top \mathbf{V})$ according to associative property, the computational complexity is reduced to $\mathcal{O}(N)$.

In practice, the attention operations in Eq. (1) and (2) can be implemented in autoregressive models, where calculation of $\mathbf{u}_t$ can only see the proceeding tokens with $j \leq t$. Specifically, with the causal masking, the linear attention can be rewritten as:

$$\mathbf{u}_t = \frac{\mathbf{Q}_t \left( \sum_{j=1}^{t} \mathbf{K}_j^\top \mathbf{V}_j \right)}{\mathbf{Q}_t \left( \sum_{j=1}^{t} \mathbf{K}_j^\top \right)} = \frac{\mathbf{Q}_t \mathbf{S}_t}{\mathbf{Q}_t \mathbf{G}_t}, \quad \mathbf{S}_t = \mathbf{S}_{t-1} + \mathbf{K}_t^\top \mathbf{V}_t, \ \mathbf{G}_t = \mathbf{G}_{t-1} + \mathbf{K}_t^\top, \quad (3)$$

where $\mathbf{S}_t = \sum_{j=1}^{t} \mathbf{K}_j^\top \mathbf{V}_j$ and $\mathbf{G}_t = \sum_{j=1}^{t} \mathbf{K}_j^\top$. This enables recurrent computation of causal linear attention by cumulatively updating $\mathbf{S}_t$ and $\mathbf{G}_t$, which serve as internal hidden states.

In the autoregressive process with causal masking, the softmax attention operation Eq. (1) in Transformer can be seen as a recurrent process based on an accumulated set of key-value pairs $\{(\mathbf{K}_j, \mathbf{V}_j)\}_{j=1}^{t}$ as a hidden state (Katharopoulos et al., 2020).

**Structured state space sequence models (SSM or S4)** (Gu & Dao, 2023; Gu et al., 2021a; Dao & Gu, 2024) are sequence models describing a system that maps input $z_t \in \mathbb{R}$ to output $u_t \in \mathbb{R}$ through a hidden state $\mathbf{h}_t \in \mathbb{R}^{C \times 1}$ in a *discrete* sequence applied with neural networks. Specifically, SSMs can be formulated with parameters $\mathbf{A} \in \mathbb{R}^{C \times C}$, $\mathbf{B} \in \mathbb{R}^{C \times 1}$, $\mathbf{C} \in \mathbb{R}^{1 \times C}$, and $D \in \mathbb{R}$, as

$$\mathbf{h}_t = \mathbf{A}\mathbf{h}_{t-1} + \mathbf{B}z_t, \ \ u_t = \mathbf{C}\mathbf{h}_t + Dz_t. \quad (4)$$

We directly formulate a discrete SSM in Eq. (4), where the $\mathbf{A}$ and $\mathbf{B}$ are transformed from a *continuous* version $\mathbf{A}'$ and $\mathbf{B}'$ relying on a timescale parameter $\Delta \in \mathbb{R}$, via $\mathbf{A} = \exp(\Delta \mathbf{A}')$ and $\mathbf{B} = \mathbf{A}^{-1}(\mathbf{A} - \mathbf{I}) \cdot \Delta \mathbf{B}'$. $\mathbf{A}$ and $\mathbf{B}$ perform the selection or gating in hidden state updating.

### 3.2 SSM AND MAMBA FOR META-CONTINUAL LEARNING

**Selective SSM & Mamba in MCL.** The dynamics of the basic SSM or S4 are time-invariant, restricting the model's ability to handle complex sequences. Mamba (Gu & Dao, 2023) incorporates a selective SSM into the model by generating the input-dependent SSM parameters to reflect the input/step-sensitive selection process. The selective SSM can be written as:

$$\mathbf{h}_t = \mathbf{A}_t \mathbf{h}_{t-1} + \mathbf{B}_t z_t, \ \ u_t = \mathbf{C}_t \mathbf{h}_t + Dz_t, \quad (5)$$

where $\mathbf{A}_t$, $\mathbf{B}_t$, and $\mathbf{C}_t$ are produced in Mamba relying on the input token at step $t$. Different from Transformers maintaining key-value pairs for all input tokens (leading to linearly increasing state size), Mamba compresses the context information in a fixed/constant-size hidden state, matching the efficiency requirements and original objective of CL.

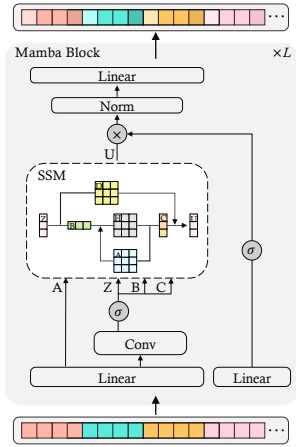

Figure 2: Illustration of the designs of Mamba block.

In our MCL tasks and other practical scenarios, we need Mamba to handle the input sequence $\mathbf{Z} \in \mathbb{R}^{N \times M}$ with each token as a vector $\mathbf{z}_t \in \mathbb{R}^M$. Mamba applies the selective SSM to each dimension/channel independently:

$$\mathbf{H}_t = [\mathbf{A}_{t,i}\mathbf{h}_{t-1,i} + \mathbf{B}_{t,i}\mathbf{z}_{t,i}]_{i=1}^{M}, \ \ \mathbf{u}_t = \mathbf{C}_t \mathbf{H}_t + \mathbf{D} \odot \mathbf{z}_t, \quad (6)$$

where $\mathbf{H}_t \in \mathbb{R}^{C \times M}$ is a concatenation of the hidden state corresponding to all $M$ dimensions of the input embedding, $\mathbf{C}_t \in \mathbb{R}^{1 \times C}$, $\mathbf{D} \in \mathbb{R}^{1 \times M}$, and $\mathbf{u}_t \in \mathbb{R}^{1 \times M}$. As shown in Fig. 2, the Mamba block used in our work applies a 1-D convolution on the input tokens and then projects the representations to obtain the input-dependent SSM parameters (Dao & Gu, 2024). Multiple Mamba blocks are stacked homogeneously. Relying on the selective mechanism (Gu & Dao, 2023), Mamba's ability to handle complex MCL tasks can be stronger than other attention-free models and competitive or better than Transformers with a key-value cache.

#### 3.2.1 MCL WITH MAMBA AS A CONTINUAL LEARNER

We will train a Mamba model $f_\theta()$ to perform CL by processing an online data stream containing paired $(\mathbf{x}, y)$; the model can produce predictions relying on the retained hidden state, for all the seen tasks. Meta-learning of such continual learner will be conducted on multiple CL episodes. Each CL episode contains a training data stream $\mathcal{D}^{\text{train}}$ and a testing set $\mathcal{D}^{\text{test}}$ from the same task distribution, denoted as $P_{(\mathcal{X}, \mathcal{Y})}$ with $(\mathcal{D}^{\text{train}}, \mathcal{D}^{\text{test}}) \sim P_{(\mathcal{X}, \mathcal{Y})}$. For example, in ICL, all classes used for testing

should have been seen in the preceding classes in the data stream. The objective of MambaCL is to meta-learn the parameter of Mamba model, *i.e.*, $\theta$, to perform prediction $\hat{y}^{\text{test}} = f_\theta((\mathcal{D}^{\text{train}}, \mathbf{x}^{\text{test}}))$ for any $(\mathbf{x}_i^{\text{test}}, \mathbf{y}_i^{\text{test}}) \in \mathcal{D}^{\text{test}}$. The CL task can be treated as next token prediction problem in sequence: $(\mathbf{x}_1^{\text{train}}, \mathbf{y}_1^{\text{train}}, ..., \mathbf{x}_T^{\text{train}}, \mathbf{y}_T^{\text{train}}, \mathbf{x}_k^{\text{test}}) \rightarrow y_k^{\text{test}}$. The meta-learning of a Mamba continual learner can be performed by optimizing the sequence prediction task on a series of sampled CL episodes:

$$\min_\theta \mathbb{E}_{(\mathcal{D}^{\text{train}}, \mathcal{D}^{\text{test}}) \sim P_{(\mathcal{X}, \mathcal{Y})}} \sum_{(\mathbf{x}^{\text{test}}, y^{\text{test}}) \in \mathcal{D}^{\text{test}}} \ell(f_\theta((\mathcal{D}^{\text{train}}, \mathbf{x}^{\text{test}})), y^{\text{test}}), \tag{7}$$

where $\ell(\cdot, \cdot)$ denotes the proper loss function for different tasks, *e.g.*, classification or regression.

On the data stream, the meta-learned Mamba $f_\theta()$ recognizes the association relationship between $\mathbf{x}$ and $y$ through the sequence, and then recurrently updates the hidden state $\mathbf{H}_t$, which can used for prediction, as shown in Fig. 1. This efficient online CL process selects and compresses the knowledge in the data stream in a time-variant and content-aware selective manner. To further validate the extension ability of Mamba in MCL, we also explore the potential of incorporating mixture-of-expert (MoE) architecture into Mamba model (Fedus et al., 2022; Pioro et al., 2024) for learning and mixing multiple learners.

**Target token embeddings.** The value of the target $y$ is essentially a symbol with consistent indication meaning for $\mathbf{x}$ within each episode, which does not take any global meaning across the episode. The model is thus trained to handle arbitrary CL episodes with the ability to generalize to different domains. Instead of pre-defining a small and fixed feasible set of candidate targets *e.g.*, classes, and a restricted prediction head, we conduct token embeddings for targets based on a universal and large vocabulary (Lee et al., 2024), inspired by the tokenization in LMs (Sennrich, 2015; Devlin et al., 2019). For each episode, a subset of unique codes is randomly picked from the vocabulary to indicate different classes; in inference, the sequence model produces the probability of the next step for all possible tokens in the vocabulary. Instead of conducting experiments of meta-training and meta-testing with the same number of classes Lee et al. (2024), we conduct generalization analyses.

### 3.2.2 REGULARIZING SELECTIVITY OF MAMBA FOR META-TRAINING

It is non-trivial to meta-learn the continual learner for associating the input and target by seeing a data stream, for both Transformers and attention-free models. The meta-training can be slow to converge or hard to find an optimal solution. We thus consider giving additional guidance in meta-training, by enhancing the association between the query tokens (*i.e.*, testing input) with correlated preceding tokens. During training, for an input $\mathbf{x}$ (corresponding to a pair $(\mathbf{x}, y)$) in the stream at the step $2t + 1$ after $2t$ tokens of $t$ samples, its association relationship with preceding tokens can be represented as $\mathbf{p}_{2t+1} = [\mathbb{1}_{y_{2t+1}}(y_1), \mathbb{1}_{y_{2t+1}}(y_1), ..., \mathbb{1}_{y_{2t+1}}(y_t), \mathbb{1}_{y_{2t+1}}(y_t)]$ with $\mathbf{p} \in \{0, 1\}^{2t}$, where $\mathbb{1}_y(y')$ is an indicator function with $\mathbb{1}_y(y') = 1$, if $y = y'$, and $\mathbb{1}_y(y') = 0$, if $y \neq y'$. We hope the meta-learned learner can also identify and use this pattern in CL (*i.e.*, meta-testing).

Transformers maintain the key-value pairs for all samples as the state. For prediction at a step, attention is applied to all the stored keys through a query, retrieving the learned information. As shown in Eq. (1), for the token at step $2t + 1$, the attention weights/patterns to previous-step tokens can be denoted as $\mathbf{q}_{2t+1}^{\text{Trans}} = [\mathbf{Q}_{2t+1}\mathbf{K}_j^\top]_{j=1}^{2t} \in \mathbb{R}^{2t}$. Note that we omit the normalization terms in attention weights to simplify the presentation. The meta-learning guidance can be applied by encouraging the similarity between $\mathbf{q}_{2t+1}^{\text{Trans}}$ and $\mathbf{p}_{2t+1}$.

Mamba and other attention-free methods (*e.g.*, Linear Transformer) compress knowledge in a hidden state at each step, as shown in Eq. (3) and (5). Specifically, Mamba applies an input-dependent selection and gating at each step. Although there are no explicit attention weights produced in Mamba, we formulate the regularization for the selectivity of Mamba by bridging the selective SSM (in Eq. (5) and (6) ) with linear attention (in Eq. (3)) and the softmax attention (in Eq. (1) ). As shown in Eq. (3), Linear Transformer updates the state $\mathbf{S}$ (and the normalization term $\mathbf{G}$) using kernel-based $\mathbf{K}$ and $\mathbf{V}$, and performs prediction based on $\mathbf{Q}$. Considering that the $\mathbf{K}$, $\mathbf{V}$, and $\mathbf{Q}$ in linear attention share the same meaning as in the softmax attention, we still can obtain $\mathbf{q}_{2t+1}^{\text{LNTrans}} = [\mathbf{Q}_{2t+1}\mathbf{K}_j^\top]_{j=1}^{2t}$ by storing the $\mathbf{K}_j$ of intermediate tokens only during training for regularization. By examining the duality relationship between the SSM in Eq. (5) and the formulation of Linear Transformer in Eq. (3) (Dao & Gu, 2024), we can identify the connections between the selective parameters, *i.e.*, $\mathbf{C}_t$ and $\mathbf{B}_t$, in SSM and query-key embeddings, *i.e.*, $\mathbf{Q}_t$ and $\mathbf{K}_t$, in linear attention. Relying on the linear attention as the bridge, we can obtain the associative indicators of Mamba as $\mathbf{q}_{2t+1}^{\text{Mamba}} = [\mathbf{C}_{2t+1}\mathbf{B}_j^\top]_{j=1}^{2t}$. To regularize the models' attention or selection behavior in meta-training,

for a query sample $(\mathbf{x}, y)$ in a sequence, we apply a selectivity regularization:

$$\ell_{\text{slct}}((\mathbf{x}, y)) = \text{KL}(\mathbf{p}_{\text{idx}((\mathbf{x},y))}, \mathbf{q}^*_{\text{idx}((\mathbf{x},y))}), \tag{8}$$

where $\text{idx}()$ indicates the step of the token $\mathbf{x}$, $*$ indicates the arbitrary model, and KL divergence is used to minimize the difference between model's association pattern and the ground truth. Note that this regularization and maintained intermediate components are not necessary in inference. We apply this regularization to MambaCL and other sequence prediction models (weighted by a scalar $\lambda$) together with the MCL objective in Eq. (7), which improves the meta-training stability and convergence for all models.

## 4 EXPERIMENTS AND ANALYSES

**Experimental setup.** To evaluate the performance of various architectures across multiple types of tasks, we conducted a series of experiments. Firstly, we divided one dataset into multiple tasks, typically with each task representing a distinct class within the dataset. We distributed these tasks into two non-overlapping sets, *i.e.*, meta-training and meta-testing. The construction of CL episodes for both meta-groups follows the same procedure: for each CL episode, we randomly select $K$ distinct tasks. $K$ is set as 20 by default. We also investigated scenarios with different of $K$ values. By default, each task in both the training and testing sequences includes five samples (5-shot). Additionally, involving fewer and more shots were also explored to further assess adaptability and learning efficiency.

**Datasets.** We conduct experiments across various datasets: **general image classification tasks** included Cifar-100 (Krizhevsky & Hinton, 2009), ImageNet-1K (Russakovsky et al., 2015), ImageNet-R (Russakovsky et al., 2015), MS-Celeb-1M (Celeb) (Guo et al., 2016), CASIA Chinese handwriting (Casia) (Liu et al., 2011), and Omniglot (Lake et al., 2015); **fine-grained recognition tasks** involved CUB-200 (Wah et al., 2011), Stanford Dogs (Khosla et al., 2011), Stanford Cars (Krause et al., 2013), and FGVC-Aircraft (Aircraft) (Maji et al., 2013); the **large domain shift tasks** featured (Peng et al., 2019); and **regression tasks** consisted of sine wave reconstruction (sine), image rotation prediction (rotation), and image completion (completion).

**Implementation details.** We conduct our main experiments on a single NVIDIA A100 GPU. We repeated each experiment five times and reported the mean and standard deviation of these runs. Results are reported upon convergence on the meta-training set. The batch size is set to 16, and the Adam optimizer is applied. We set the initial learning rate to $1 \times 10^{-4}$, with decays of 0.5 every 10,000 steps. For all models, we ensure a consistent setup to enable fair comparisons and make sure all models achieve satisfactory results, with additional details provided in Sec. B. Specifically, for experiments involving training from scratch, we adopt the settings from (Lee et al., 2024) to maintain fairness. For the networks built on pre-trained models, we use the OpenAI/CLIP-ViT-B16 (Radford et al., 2021; Ilharco et al., 2021) as our image encoder, with its parameters frozen during training and an additional trainable linear projector.

### 4.1 EXPERIMENTAL RESULTS AND ANALYSES

In our experiments, we assess several models including OML (Javed & White, 2019), Vanilla Transformer (Vaswani et al., 2017), Linear Transformer (Katharopoulos et al., 2020), Performer (Choromanski et al., 2020), and our MambaCL. OML serves as a conventional SGD-based meta-continual learning baseline, featuring a two-layer MLP prediction network on top of a meta-learned encoder. Transformers exhibit advanced sequence modeling capabilities, but they may not be optimal for (CL) due to computational inefficiencies and the broad objectives associated with CL. To enhance efficiency, Linear Transformer and Performer utilize kernel methods and linear operations to approximate softmax attention, which maintain a constant hidden state size and exhibit linear computational complexity. All transformer models share a similar structure, each with 4 layers and 512 hidden dimensions. Mamba is an attention-free model optimized for efficiently processing long sequences with near-linear computational demands. Our Mamba Learner also utilizes 4 layers and 512 hidden dimensions, facilitating comparison with the transformer models, yet it features significantly fewer parameters.

**General image classification tasks.** Table 1 and 2 present comparative performance analyses of different architectures on several general image classification tasks, initiating training from scratch and extracting image representation based on a pre-trained model, respectively. In Table 1, within the CIFAR-100 datasets, all methods suffer from substantial meta-overfitting, as evidenced by the large gap between meta-training and meta-testing scores. This may be attributed to the lower task (class)

Table 1: Classification accuracy (%) across 20-task 5-shot MCL, training from the scratch on general image classification tasks. The **best** and **second best** performances are indicated in **red** and **blue**, respectively.

| Method | Cifar-100 | | Omniglot | | Casia | | Celeb | |
|---|---|---|---|---|---|---|---|---|
| | Meta-Train | Meta-Test | Meta-Train | Meta-Test | Meta-Train | Meta-Test | Meta-Train | Meta-Test |
| OML | $99.4^{\pm0.1}$ | $10.1^{\pm0.4}$ | $99.9^{\pm0.0}$ | $75.2^{\pm2.2}$ | $97.2^{\pm0.1}$ | $96.8^{\pm0.1}$ | $58.2^{\pm0.3}$ | $57.5^{\pm0.2}$ |
| Transformer | $100.0^{\pm0.0}$ | $17.2^{\pm0.8}$ | $100.0^{\pm0.0}$ | $86.3^{\pm0.6}$ | $99.7^{\pm0.0}$ | $99.6^{\pm0.0}$ | $70.9^{\pm0.2}$ | $70.0^{\pm0.2}$ |
| Linear TF | $99.9^{\pm0.1}$ | $16.6^{\pm0.5}$ | $100.0^{\pm0.0}$ | $64.0^{\pm1.4}$ | $99.6^{\pm0.0}$ | $99.3^{\pm0.0}$ | $68.9^{\pm0.3}$ | $67.6^{\pm0.3}$ |
| Performer | $100.0^{\pm0.0}$ | $17.1^{\pm0.3}$ | $99.9^{\pm0.1}$ | $62.9^{\pm4.6}$ | $99.5^{\pm0.0}$ | $99.3^{\pm0.0}$ | $67.5^{\pm0.5}$ | $66.3^{\pm0.2}$ |
| Mamba | $99.9^{\pm0.1}$ | $18.3^{\pm0.4}$ | $100.0^{\pm0.0}$ | $87.7^{\pm0.5}$ | $99.8^{\pm0.1}$ | $99.5^{\pm0.1}$ | $69.4^{\pm0.2}$ | $68.1^{\pm0.1}$ |

Table 2: Classification accuracy (%) across 20-task 5-shot MCL, training from the pre-trained models on general image classification tasks.

| Method | Cifar-100 | ImageNet-1K | ImageNet-R | Celeb | Casia | Omniglot |
|---|---|---|---|---|---|---|
| OML | $64.4^{\pm0.4}$ | $90.5^{\pm0.3}$ | $67.5^{\pm0.3}$ | $72.8^{\pm0.1}$ | $81.5^{\pm0.5}$ | $90.4^{\pm0.2}$ |
| Transformer | $62.7^{\pm0.7}$ | $93.5^{\pm0.1}$ | $63.6^{\pm0.2}$ | $78.4^{\pm0.1}$ | $93.8^{\pm0.2}$ | $94.4^{\pm0.2}$ |
| Linear TF | $54.3^{\pm0.7}$ | $89.1^{\pm0.2}$ | $55.7^{\pm0.3}$ | $76.5^{\pm0.2}$ | $90.9^{\pm0.4}$ | $86.5^{\pm0.5}$ |
| Performer | $53.4^{\pm0.3}$ | $90.8^{\pm0.5}$ | $52.8^{\pm0.9}$ | $76.8^{\pm0.1}$ | $93.0^{\pm0.3}$ | $89.3^{\pm0.3}$ |
| Mamba | $67.1^{\pm0.4}$ | $93.6^{\pm0.2}$ | $69.7^{\pm0.4}$ | $77.0^{\pm0.1}$ | $93.1^{\pm0.2}$ | $95.9^{\pm0.2}$ |

diversity. In Table 2, the results for continual learners built on pre-trained models exhibit similar trends in CIFAR-100 and ImageNet-R. Furthermore, Mamba demonstrates superior performance compared to other methods in these scenarios, underscoring its robustness against overfitting. On larger datasets such as ImageNet-1K, Casia, and Celeb, Mamba performs on par with or surpasses transformers. Without losing generality, we use the pre-trained image representations for our experiments by default.

Table 3: Classification accuracy (%) across 20-task 5-shot MCL on fine-grained recognition tasks.

| Method | CUB-200 | Dogs | Cars | Aircraft |
|---|---|---|---|---|
| OML | $78.7^{\pm0.6}$ | $72.4^{\pm0.5}$ | $83.6^{\pm0.7}$ | $49.5^{\pm0.2}$ |
| Transformer | $81.4^{\pm0.4}$ | $77.5^{\pm0.6}$ | $87.0^{\pm0.3}$ | $53.9^{\pm0.7}$ |
| Linear TF | $69.7^{\pm0.7}$ | $69.7^{\pm0.7}$ | $76.6^{\pm0.8}$ | $49.0^{\pm0.7}$ |
| Performer | $69.2^{\pm0.8}$ | $69.4^{\pm0.4}$ | $73.9^{\pm0.8}$ | $48.6^{\pm0.6}$ |
| Mamba | $83.0^{\pm0.4}$ | $79.2^{\pm0.5}$ | $88.3^{\pm0.4}$ | $55.3^{\pm0.6}$ |

**Fine-grained recognition tasks.** Table 3 presents a performance comparison of different architectures on fine-grained recognition datasets. In fine-grained datasets, where only subtle differences exist between classes (*e.g.*, the CUB-200 dataset, which contains 200 bird subcategories), models need capture global information across the entire training episode to distinguish these fine-grained differences. Mamba outperforms other models across these datasets, potentially due to its robustness to capture subtle inter-class distinctions.

### 4.2 GENERALIZATION ANALYSES

We hope a meta-learned learner has the ability to be generalized to unseen scenarios. We conduct generalization analyses for Transformer models and Mamba in scenarios involving generalization to longer untrained sequence lengths, larger domain shifts, and sensitivity to the noise inputs, for meta-testing. Additionally, to analyze the behaviors of these models, we visualize the attention weights of Transformers and the associative weights of Mamba to demonstrate their attentions and selectivity patterns in Sec. D.

**Generalization to different stream length.** To effectively address episodes of continual learning of indefinite length, the learning algorithm should demonstrate the capability to generalize beyond the sequence lengths observed during meta-training. We conducted length generalization experiments on ImageNet-1K, training vanilla Transformers, linear Transformers, and Mamba on 20-task 5-shot MCL, each with a vocabulary of 200 tokens. The length of a continual learning episode is calculated as $2 \times \text{tasks} \times \text{shots} + 1$.

*Meta-testing on different numbers of tasks.* Fig. 3a shows the performance of the three models meta-trained on a 20-task, 5-shot setup, evaluated during meta-testing across varying numbers of tasks while keeping a constant shot number of 5. Both the Transformer and Linear Transformer suffer significant performance degradation when meta-testing at untrained episode lengths, even for

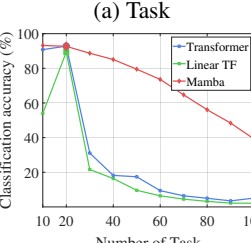 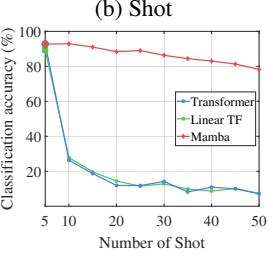 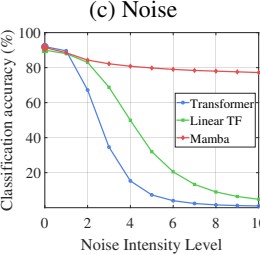

Figure 3: Generalization Analysis on ImageNet-1K: (a) meta-trained on 20-task 5-shot MCL, meta-testing on varying number of tasks (5-shot); (b) meta-trained on 20-task 5-shot MCL, meta-testing on varying number of shots (20-task); (c) varying inputs noise intensity level.

simpler tasks such as the 10-task, 5-shot configuration. Mamba's meta-testing performance on the 10-task setup is better relative to the meta-trained 20-task setup, and the performance degradation is relatively mild compared to transformers as the number of tasks gradually increases.

*Meta-testing on different number of shots.* In Fig. 3b, we evaluate the performance of three models meta-trained on a 20-task, 5-shot setup, evaluated during meta-testing across varying numbers of shots while maintaining a constant task count of 20. Both the vanilla Transformer and linear Transformer exhibit significant performance degradation, likely due to overfitting the 20-task, 5-shot pattern. However, Mamba experiences only about a 10% performance degradation when the meta-testing shot number reaches 50, which is ten times the meta-training episode length. Fig. 3a and 3b demonstrate Mamba's robustness in length generalization.

**Results and analyses on larger domain shift.** We explore a larger domain shift scenario using the DomainNet dataset (containing 6 different domains) to further evaluate model generalization to unseen input distributions, with one domain reserved for meta-testing and the remaining domains for meta-training, which represents a more realistic setting. The experimental results are presented in Table 4. Overall, these models demonstrate the capability to handle large domain shift scenarios. Mamba performs on par with or surpasses Transformer models across various target domains, benefiting from the potentially better generalization ability from smaller-size model with less overfitting possibility. Vanilla Transformers perform well when the targets are real images or paintings. Mamba excels particularly in the Quickdraw domain, which exhibits larger differences compared to other domains. This performance may be attributed to Mamba's robustness in processing inputs with larger deviations from the training distribution.

Table 4: Classification accuracy (%) across 20-task 5-shot MCL on DomainNet dataset. (*inf,pnt,qdr,rel,skt→clp* denotes meta-testing on Clipart domain, and the remaining domains used for meta-training. *clp: clipart, inf: infograph, pnt: painting, qdr: quickdraw, rel: real, skt: sketch.*)

| Method | inf,pnt,qdr, rel,skt→clp | clp,pnt,qdr, rel,skt→inf | clp,inf,qdr, rel,skt→pnt | clp,inf,pnt, rel,skt→qdr | clp,inf,pnt, qdr,skt→rel | clp,inf,pnt, qdr,rel→skt | Avg |
|---|---|---|---|---|---|---|---|
| Transformer | $91.8^{\pm0.1}$ | $69.4^{\pm0.1}$ | $82.6^{\pm0.2}$ | $50.2^{\pm0.6}$ | $93.8^{\pm0.1}$ | $85.9^{\pm0.3}$ | $79.0^{\pm0.3}$ |
| Linear TF | $91.0^{\pm0.0}$ | $66.2^{\pm0.8}$ | $80.6^{\pm0.8}$ | $30.7^{\pm1.4}$ | $92.9^{\pm0.1}$ | $85.5^{\pm0.1}$ | $74.5^{\pm0.5}$ |
| Performer | $91.3^{\pm0.2}$ | $66.4^{\pm0.6}$ | $81.3^{\pm0.2}$ | $39.4^{\pm1.7}$ | $92.8^{\pm0.1}$ | $84.8^{\pm0.5}$ | $76.0^{\pm0.6}$ |
| Mamba | $91.7^{\pm0.2}$ | $70.2^{\pm0.2}$ | $81.8^{\pm0.2}$ | $55.6^{\pm0.8}$ | $93.0^{\pm0.1}$ | $87.2^{\pm0.3}$ | $79.9^{\pm0.3}$ |

**Sensitivity to the noisy inputs.** To evaluate the sensitivity of different models to noisy inputs, we conduct experiments on meta-trained 20-task, 5-shot MCL models using the ImageNet-1K dataset. Within each meta-testing episode, we apply noise to the input embeddings $\mathbf{x}_i$ of *five* randomly selected samples. We add the noise following Gaussian distributions characterized by a mean ($\mu$) of 0 and a standard deviation ($\sigma$) that ranges from 0 to 10. As depicted in Fig. 3c, the vanilla transformer and linear transformer suffer significant performance degradation. In contrast, Mamba demonstrates robust performance when processing inputs with high levels of noise.

## 4.3 TRANING ON LONGER EPISODES

We conducted experiments to meta-train the models on longer episodes across both classification and regression tasks. Table 5 demonstrates that Mamba continues to perform comparably to Transformer, and significantly outperforms SGD-based approaches (OML).

Table 5: Classification accuracy (%) and regression errors across 100-task 5-shot MCL.

| Method | Casia | Celeb | Sine | Rotation | Completion |
|---|---|---|---|---|---|
| OML | $93.2^{\pm0.9}$ | $45.5^{\pm0.2}$ | $0.0498^{\pm0.0004}$ | $0.524^{\pm0.087}$ | $0.1087^{\pm0.0001}$ |
| Transformer | $99.0^{\pm0.0}$ | $60.5^{\pm0.1}$ | $0.0031^{\pm0.0002}$ | $0.031^{\pm0.001}$ | $0.0989^{\pm0.0001}$ |
| Linear TF | $97.7^{\pm0.1}$ | $54.7^{\pm0.1}$ | $0.0139^{\pm0.0003}$ | $0.047^{\pm0.002}$ | $0.1084^{\pm0.0001}$ |
| Mamba | $99.1^{\pm0.1}$ | $59.9^{\pm0.1}$ | $0.0054^{\pm0.0001}$ | $0.025^{\pm0.001}$ | $0.0895^{\pm0.0001}$ |

## 4.4 ABLATION STUDIES

**Hyper-parameter of selectivity regularization loss.** We conducted an ablation study to assess the influence of training loss hyper-parameter on our Mamba model's efficacy. Specifically, this study involved adjusting the $\lambda$ values within our selectivity regularization loss, experimenting with hyper-parameters set at $\{0.1, 0.2, 0.5, 1.0, 2.0\}$, as depicted in Fig. 4. The results indicate that these variations have a minor impact on our Mamba model's performance. Consequently, we selected a $\lambda$ value of $0.5$ for our experiments.

**SSM state size.** In Fig. 5, we evaluate the impact of varying the SSM state size on the performance of our methods. We conducted experiments on ImageNet-1K and Cifar-100, training MambaCL with state sizes of 16, 32, 64, 128, and 256. The results show consistent performance improvement as state size increases. To balance performance and computational cost, we set the state size as 128.

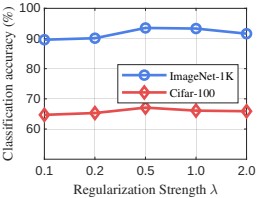

Figure 4: Ablation of varying $\lambda$ in training loss.

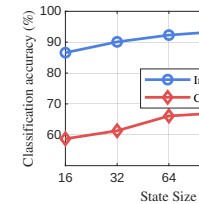

Figure 5: Ablation of varying SSM state size.

Table 6: Different Mamba architectures on 20-tasks 5-shot MCL.

| Method | Cifar-100 | ImageNet-1K |
|---|---|---|
| Transformer | $62.7^{\pm0.7}$ | $93.5^{\pm0.1}$ |
| Linear TF | $54.3^{\pm0.7}$ | $89.1^{\pm0.2}$ |
| Mamba-1 | $59.7^{\pm0.5}$ | $90.1^{\pm0.3}$ |
| MambaFormer | $62.4^{\pm0.6}$ | $92.7^{\pm0.1}$ |
| Mamba-2 | $67.1^{\pm0.4}$ | $93.6^{\pm0.2}$ |
| Mamba+MoE | $68.9^{\pm0.2}$ | $94.0^{\pm0.2}$ |

**Different architectures.** In Table 6, we present an ablation study comparing different Mamba architectures in our MambaCL, including Mamba-1, MambaFormer (Park et al., 2024), and Mamba-2. MambaFormer is a hybrid model that integrates the vanilla attention mechanism of Mamba-1 and replaces the transformer's positional encoding with a Mamba block. The results in Table 6 demonstrate that MambaFormer also achieved performance comparable to that of the transformer. However, Mamba-2 performed better on Cifar-100 than the other variants.

**Mamba+MoE.** In Table 6, we present experiments where Mamba was enhanced with Mixture of Experts (MoE), incorporating twelve 2-layer MLP expert networks with a dense-MoE router following each Mamba Block, resulting in improved performance. Additionally, we include performance metrics for vanilla and linear transformers for reference.

Table 7: Computational cost on 20-task 5-shot MCL.

| Methods | Params.↓ | Inf. Speed↑ |
|---|---|---|
| TF | 9.2M | 325ep/s |
| Mamba | 5.4M | 858ep/s |

**Computational cost.** In Table 7, we detail various aspects of computational cost using our implementation in PyTorch [27], executed on an NVIDIA 4090 GPU and an INTEL I9-14900k CPU. We specifically report the costs associated with meta-testing at a batch size of 1. Notably, Mamba, characterized by fewer parameters and increased processing speed, achieves performance that either matches or surpasses that of the vanilla transformer.

## 5 CONCLUSION

In this paper, we tried to answer a question – Can attention-free Mamba perform well on MCL? We formulate the SSM and Mamba as a sequence prediction-based continual learner and meta-learn it on CL episodes. A selectivity regularization is introduced for meta-learning the models. Comprehensive experiments show that Mamba performs well across diverse MCL scenarios, significantly outperforming attention-free methods and matching or exceeding Transformers' performance with fewer parameters and computations. In challenging scenarios with global structures, domain shifts, and long sequences, Mamba demonstrates obvious reliability, generalization, and robustness.

**Limitations and future work.** This study can be extended to larger-scale datasets and offline CL settings. Beyond the current MCL framework, we aim to explore the online meta-continual learning paradigm to broaden the applicability of our approach to a wider range of scenarios.

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

# A  DATASETS

## A.1  GENERAL IMAGE CLASSIFICATION TASKS

**Cifar-100** (Krizhevsky & Hinton, 2009) dataset consists of 60,000 images across 100 classes, each with 600 images. We select 60 classes at random for meta-training and use the remaining 40 for meta-testing.

**ImageNet-1K** (Russakovsky et al., 2015) dataset comprises over one million labeled images distributed across 1,000 categories. We select 600 classes at random for meta-training and use the remaining 400 for meta-testing.

**ImageNet-R**(endition) (Russakovsky et al., 2015) extends 200 ImageNet classes with a compilation of 30,000 images tailored for robustness research.

**Celeb** (Guo et al., 2016) is a large-scale facial image collection featuring approximately 10 million images of 100,000 celebrities. We randomly allocated 1,000 classes for meta-testing and assigned the remaining classes to meta-training.

**Casia Chinese handwriting** (Liu et al., 2011) dataset encompasses a total of 7,356 character classes with 3.9 million images. We randomly selected 1,000 classes for the meta-testing and allocated the remaining classes for meta-training.

**Omniglot** (Lake et al., 2015) is a collection of 1,632 handwritten characters from 50 different alphabets. The meta-training set comprises 963 classes, while the meta-testing set includes 660 classes, with each class containing 20 images.

## A.2  FINE-GRAINED RECOGNITION TASKS

**CUB-200-2011** (Wah et al., 2011) is a widely used fine-grained visual categorization dataset, comprising 11,788 images across 200 bird subcategories. We randomly selected 80 classes for the meta-testing and allocated the remaining classes for meta-training.

**Stanford Dogs** (Khosla et al., 2011) dataset comprises 20,580 images spanning 120 global dog breeds, divided into 12,000 training images and 8,580 testing images. We select 48 classes at random for meta-testing and use the remaining 72 for meta-training.

**Stanford Cars** (Krause et al., 2013) comprises 16,185 images across 196 car classes, primarily captured from the rear perspective. We select 80 classes at random for meta-testing and use the remaining 40 for meta-training.

**FGVC-Aircraft** (Maji et al., 2013) dataset comprises 10,200 images across 102 aircraft model variants, each represented by 100 images, primarily consisting of airplanes. We randomly selected 40 classes for the meta-testing and allocated the remaining classes for meta-training.

## A.3  LARGE DOMAIN SHIFT TASKS

**DomainNet** (Peng et al., 2019) dataset is a benchmark for domain adaptation, encompassing common objects organized into 345 classes across six domains: clipart, real, sketch, infograph, painting, and quickdraw. We evaluate model adaptability to out-of-domain data by using one domain for meta-testing and the remaining domains for meta-training.

### A.3.1  REGRESSION TASKS

**Sine Wave Reconstruction (Sine)** The sine wave $\omega(\tau) = A\sin(2\pi\nu\tau + \psi)$ is defined by its amplitude $A$, frequency $\nu$, and phase $\psi$. We denote the target values $y$ as evaluations of the sine wave at 50 predefined points: $y = [\omega(\tau_1), \ldots, \omega(\tau_{50})]$. In each task, the frequency and phase remain constant, but the amplitude is allowed to vary. To corrupt $y$ into $x$, we introduce a phase shift and Gaussian noise, where the phase shift is randomly selected for each task. The mean squared error between $y$ and the model's prediction $\hat{y}$ is reported as the evaluation criterion.

**Image Rotation Prediction (Rotation)** The model is provided with an image rotated by an angle $\psi \in [0, 2\pi)$, and its task is to predict the rotation angle $\hat{\psi}$. We use $1 - \cos(\hat{\psi} - \psi)$ as the evaluation

Table 8: Model Configurations

|  | Mamba | Transformer | Linear TF | Performer |
|---|---|---|---|---|
| Batch size | | 16 | | |
| Max Train Step | | 50000 | | |
| Optimizer | | Adam | | |
| Learning Rate | | $1 \times 10^{-4}$ | | |
| Learning Rate Decay | | Step | | |
| Learning Rate Decay Step | | 10000 | | |
| Learning Rate Decay Rate | | 0.5 | | |
| Regularization $\lambda$ | | 0.5 | | |
| Hidden Dimension | | 512 | | |
| Layer | | 4 | | |
| State Size | 128 | - | - | - |
| Delta Convolution | 4 | - | - | - |
| Attention | - | Softmax | Elu | Favor |

metric, where a perfect prediction would result in a score of 0, while random guessing would yield an average score of 1.0. The Casia dataset is employed, with each class being treated as an individual task, maintaining the same meta-split configuration.

**Image Completion (Completion)** In this task, the model is tasked with filling in the missing parts of an image given the visible sections. Using the Casia dataset, we modify the input $x$ to consist of the top half of the image, while the target $y$ is the bottom half. The performance is evaluated by computing the mean squared error between the predicted and true pixel values. We report the MSE between $y$ and the model's prediction $\hat{y}$ as the evaluation criterion.

## B  ADDITIONAL EXPERIMENTAL DETAILS

Table 8 presents the configurations of the models employed in our experiments.

## C  ADDITIONAL EXPERIMENTS

### C.1  EFFECTS OF SELECTIVITY REGULARIZATION AND META-TRAINING LOSS CURVES

Due to the complexity of the MCL task, the proposed regularization technique plays a crucial role in stabilizing and improving the training process for all models. Fig. 6 showing the initial training phases (2500 steps) for different models with and without selectivity regularization. The losses are 3–5 times higher compared to the models with regularization applied and successfully converging. Beyond 2500 steps, the losses oscillate and no longer decrease. The results indicate that models without our regularization struggle to converge and exhibit significant oscillations during training, highlighting the effectiveness of the regularization.

### C.2  MORE ABLATION STUDIES ON REGULARIZATION STRENGTH

In Fig. 4, we conducted ablation study to assess the influence of regularization strengths on our Mamba's efficacy. Fig. 7 illustrates more ablation studies assessing the impact of regularization strengths $\lambda$ by setting is as $\{0.1, 0.2, 0.5, 1.0, 2.0\}$, across multiple models on both ImageNet-1K and Cifar-100 datasets. The results demonstrate that all models exhibit stability within a wide and appropriate range of $\lambda$, providing evidence of consistent patterns. In our experiments, without losing generality, all models employed a regularization strength of 0.5 by default.

### C.3  MORE ABLATION STUDIES ON LEARNING RATES

Fig. 8 illustrates ablation studies assessing the impact of varying initial learning rates $\{5 \times 10^{-5}, 1 \times 10^{-4}, 2 \times 10^{-4}, 5 \times 10^{-4}\}$, across multiple models on both ImageNet-1K and Cifar-100 datasets.

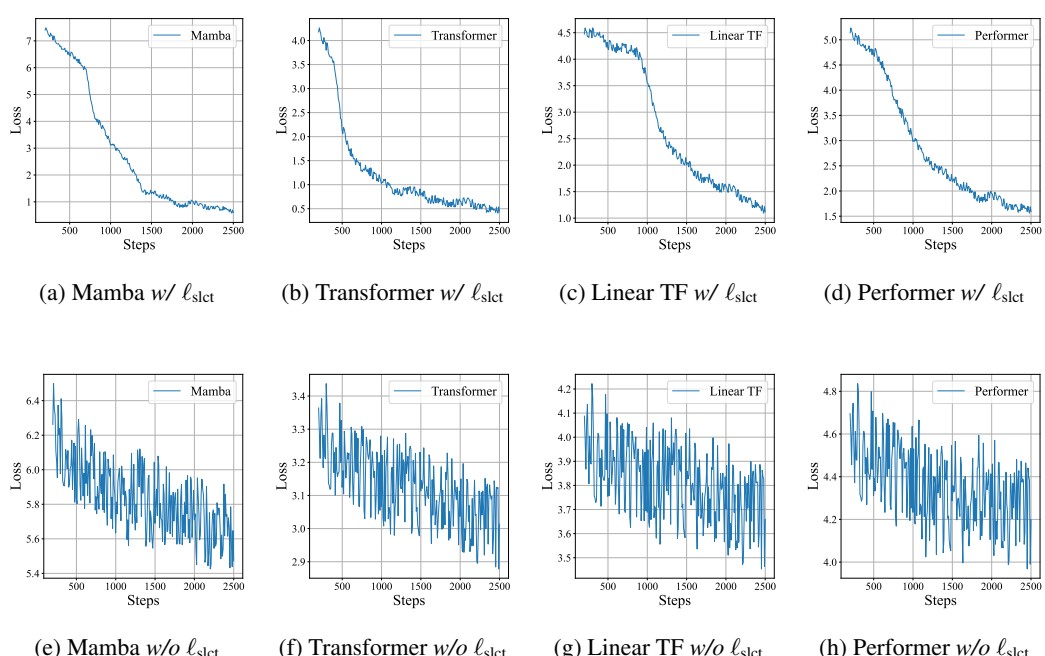

Figure 6: Training loss curves for (a, e) Mamba, (b, f) Transformer, (c, g) Linear Transformer, and (d, h) Performer, under the same type of representation and experimental settings, with and without selectivity regularization ($\ell_{\text{slct}}$) during meta-training on 20-task, 5-shot MCL on Cifar-100.

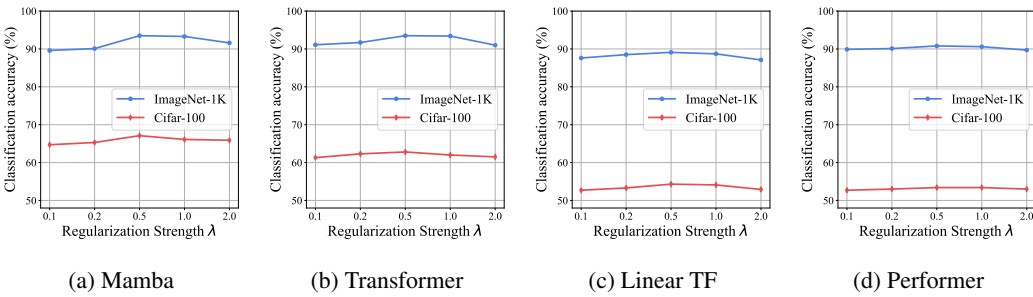

Figure 7: Ablation studies on regularization strength $\lambda$ $(0.1, 0.2, 0.5, 1.0, 2.0)$ during meta-testing of 20-task, 5-shot models (meta-trained on 20-task, 5-shot) for (a) Mamba, (b) Transformer, (c) Linear Transformer, and (d) Performer.

The results indicate that within a reasonable range, the learning rate does not significantly affect model performance. In our experiments, without losing generality, we set the initial learning rate to $1 \times 10^{-4}$, with decays of 0.5 every 10,000 steps.

## C.4 ADDITIONAL GENERALIZATION ANALYSES

Without the regularization, models struggle to converge and exhibit significant oscillations during training, as shown in Fig. 6. In Sec. 4.2 and Fig. 3, we conducted generalization analyses of various models by conducting meta-testing on the episodes different from the meta-training settings. Specifically, we apply the models meta-trained with 20-task-5-shot episodes on the meta-testing episodes with varying numbers of tasks or shots or the episodes contaminated by noise. The results show that Mamba shows better generalization ability to unseen scenarios and Transformer shows more meta-overfitting issues. To validate that the results are not relevant to the regularization, we evaluated various models with a small regularization strength ($\lambda = 0.1$) to assess the impact of regularization on this generalization experiment and the meta-overfitting issue. The results indicate that

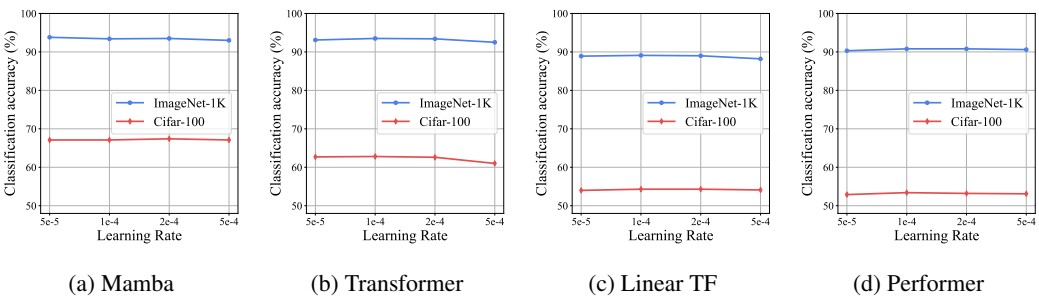

(a) Mamba      (b) Transformer      (c) Linear TF      (d) Performer

Figure 8: Ablation studies on learning rates ($\{5\times10^{-5}, 1\times10^{-4}, 2\times10^{-4}, 5\times10^{-4}\}$) during meta-testing of 20-task, 5-shot models (meta-trained on 20-task, 5-shot) for (a) Mamba, (b) Transformer, (c) Linear Transformer, and (d) Performer.

regularization strengths of 0.1 (Fig. 9) and 0.5 (Fig. 3) lead to similar phenomena across different models.

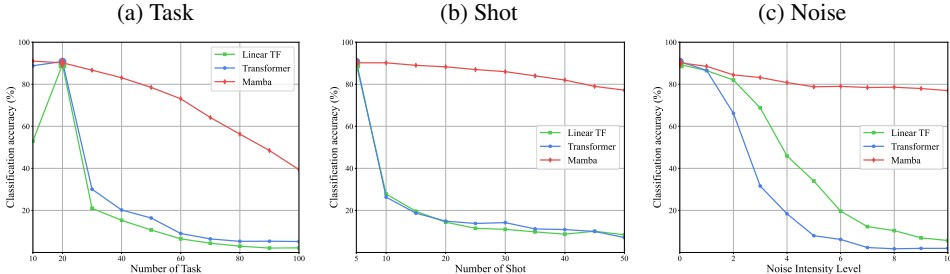

Figure 9: Generalization Analysis on ImageNet-1K with regularization strength $\lambda = 0.1$: (a) meta-trained on 20-task 5-shot MCL, meta-testing on varying number of tasks (5-shot); (b) meta-trained on 20-task 5-shot MCL, meta-testing on varying number of shots (20-task); (c) meta-trained on 20-task 5-shot MCL, meta-testing on 20-task 5-shot with varying inputs noise intensity level

## D VISUALIZATION OF ATTENTION AND SELECTIVITY PATTERN

Given meta-learned sequence models as the continual learner, the models process the samples in sequence in the meta-test CL process. To analyze the behaviors of these models, we visualize the attention weights of Transformers and the associative weights of Mamba (as discussed in Sec. 3.2.2) to demonstrate their attention and selectivity patterns, respectively. In a meta-testing episode, given a trained model and a sequence of samples, the prediction for a given $\mathbf{x}^{\text{test}}$ is produced based on the attention or implicit association of seen samples in the sequence. Visualizing the attention and selectivity patterns can empirically show how the models make predictions. For the standard benchmarking case, Fig. 10 shows that both Transformer and Mamba can effectively associate seen samples with query inputs, leading to the results as shown in Table 2.

Specifically, we use this visualization to analyze how different models perform in the generalization studies (discussed in Sec. 4.2), *i.e.*, generalizing to meta-testing cases that are different from meta-training cases.

### D.1 VISUALIZATION ANALYSES FOR GENERALIZATION TO DIFFERENT STREAM LENGTH

The experiments shown in Fig. 3a and Fig. 3b validate the generalization ability of models by meta-testing on CL episodes/sequences that differ from those seen during meta-training. Specifically, the models are meta-trained on 20-task, 5-shot MCL episodes and meta-tested on episodes with task and shot numbers exceeding those in meta-training. Transformers generally converge more easily during

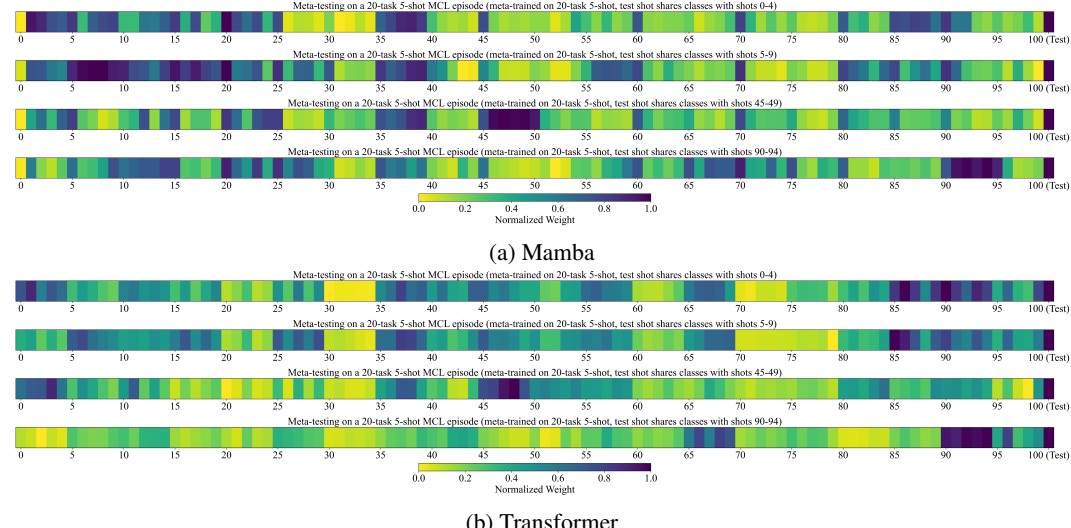

(a) Mamba

(b) Transformer

Figure 10: 20-task 5-shot in meta-testing: visualization of the final layer associations between various test shots (queries) and a single MCL train episode (prompt) of both (a) Mamba and (b) Transformer during meta-testing on 20-task 5-shot MCL episode (meta-trained on 20-task 5-shot). In meta-testing, the four visualizations share a single MCL training episode (prompt) spanning $0^{th}$–$99^{th}$ shots, while the test shots (queries at the $100^{th}$ shot) correspond to the $0^{th}$, $1^{st}$, $9^{th}$, and $18^{th}$ tasks ($0^{th}$–$4^{th}$, $5^{th}$–$9^{th}$, $45^{th}$–$49^{th}$, and $90^{th}$–$94^{th}$ train shots), respectively.

meta-training compared to Mamba, due to their strong fitting ability. However, this advantage may also lead to meta-overfitting.

To analyze how different models perform on these sequences, we visualize the final layer attention weights of Transformer and the corresponding selective scores (associative indicators) of Mamba, between various test shots (queries) and a single MCL train episode (prompt) of both Mamba and Transformer. Note that Mamba does not have explicit attention weights, we compute the scores relying on the connection between Mamba and Transformers described in Section 3.2.2. Specifically, we computed the parameters $\mathbf{C}_{test}$ and $\mathbf{B}$ ($\mathbf{C}_{test}\mathbf{B}^{\top}$) within its SSMs to compare its behavior with the attention matrix ($\mathbf{Q}_{test}\mathbf{K}^{\top}$) of Transformers, where $\mathbf{C}_{test} \in \mathbb{R}^{1 \times C}$ and $\mathbf{Q}_{test} \in \mathbb{R}^{1 \times C}$ correspond to the row of the test shot. Both models are meta-trained on a 20-task, 5-shot setting using the ImageNet-1K dataset.

For models meta-trained on the 20-task, 5-shot setting, we meta-tested them and visualized their weights on 20-task, 5-shot episodes (Fig. 10), 20-task, 10-shot episodes (Fig. 11), and 40-task, 5-shot episodes (Fig. 12). Specifically, we observed that Transformers tend to either average attention or consistently focus on specific token positions in episodes that deviate from the training length. In contrast, Mamba effectively associates with relevant shots. This suggests that Transformers may learn pattern biases in the sequences (e.g., positional biases unrelated to content), leading to meta-overfitting during these generalization tests.

## D.2 VISUALIZATION ANALYSIS OF GENERALIZATION TO NOISE-CONTAMINATED EPISODES

In the experiments, the modes are meta-trained on noise-free episodes. And the noise is added on randomly selected samples/shots in the meta-testing episodes. The task can also be seen as validating the ability of ignoring the irrelevant samples or contaminated outlier samples in the sequences. To directly show how the models work in this scenarios, we visualized the final layer attention weights for test shots compared to training shots for both Mamba and Transformer, each meta-trained in a 20-task, 5-shot setting. During meta-testing, these models processed a 20-task, 5-shot episode with five noisy input shots (shot index: 8, 18, 39, 61, 75) at noise strengths of 1 (Fig. 13), 2 (Fig. 14), and 6 (Fig. 15). The results indicate that the Transformer meta-trained on clean episodes tend to produce extreme attention weights (either very high or very low) on noisy or outlier shots,

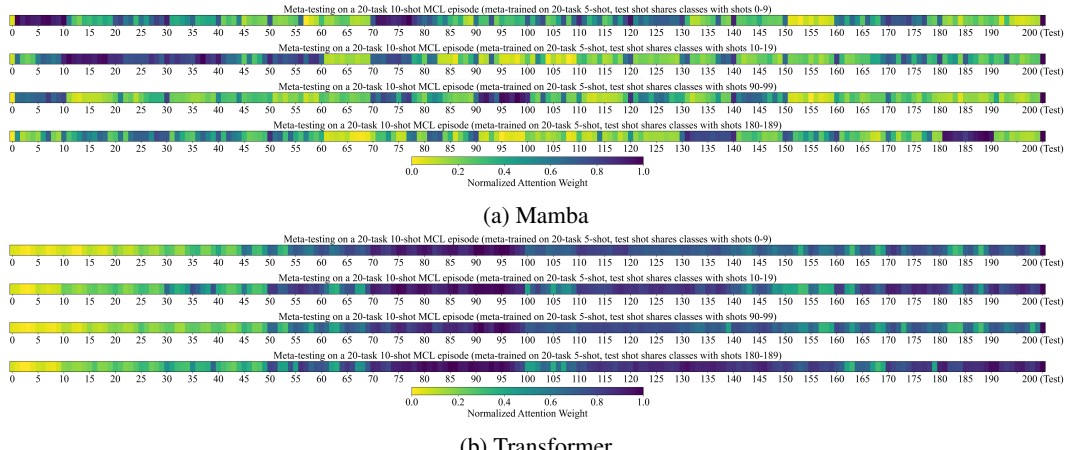

Figure 11: More shots in meta-testing: visualization of the final layer associations between various test shots (queries) and a single MCL train episode (prompt) of both (a) Mamba and (b) Transformer during meta-testing on 20-task 10-shot MCL episode (meta-trained on 20-task 5-shot). In meta-testing, the four visualizations share a single MCL training episode (prompt) spanning $0^{th}-199^{th}$ shots, while the test shots (queries at the $100^{th}$ shot) correspond to the $0^{th}$, $1^{st}$, $9^{th}$, and $18^{th}$ tasks , respectively.

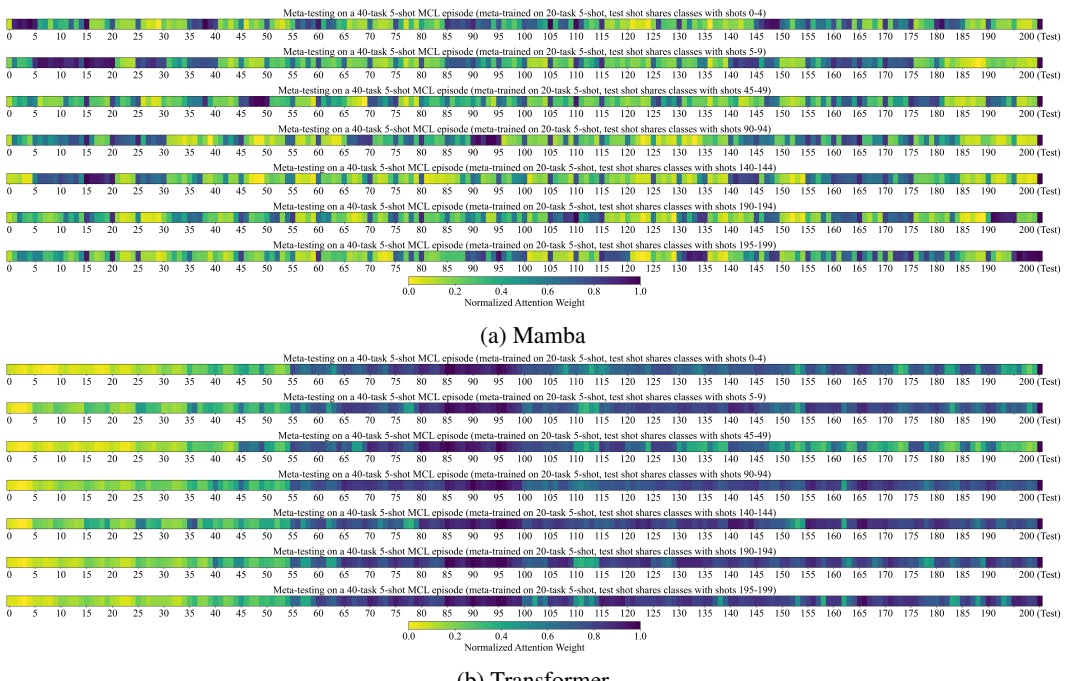

Figure 12: More tasks in meta-testing: visualization of the final layer associations between various test shots (queries) and a single MCL train episode (prompt) of both (a) Mamba and (b) Transformer during meta-testing on 40-task 5-shot MCL episode (meta-trained on 20-task 5-shot). In meta-testing, the seven visualizations share a single MCL training episode (prompt) spanning $0^{th}-199^{th}$ shots, while the test shots (queries at the $100^{th}$ shot) correspond to the $0^{th}$, $1^{st}$, $9^{th}$, $18^{th}$, $28^{th}$, $38^{th}$ and $39^{th}$ tasks , respectively.

whereas Mamba is less affected. This observation suggests that Transformers' learned attention mechanisms tend to associate samples based on local and independent representations. In contrast,

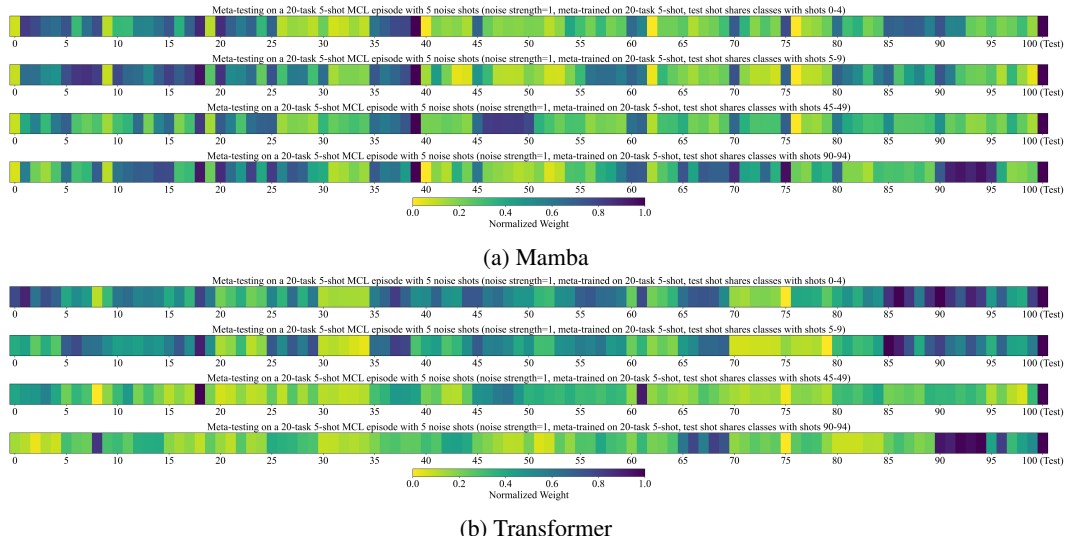

(a) Mamba

(b) Transformer

Figure 13: Noise inputs in meta-testing: visualization of the final layer associations between various test shots (queries) and a single MCL train episode (prompt) of both (a) Mamba and (b) Transformer (meta-trained on 20-task 5-shot without noise inputs), during meta-testing on 20-task 5-shot MCL episode with noise inputs (noise strength=1, noise on $8^{th}$, $18^{st}$, $39^{th}$, $61^{th}$, and $75^{th}$ training shots). In meta-testing, the four visualizations share a single MCL training episode (prompt) spanning $0^{th}$–$99^{th}$ shots, while the test shots (queries at the $100^{th}$ shot) correspond to the $0^{th}$, $1^{st}$, $9^{th}$, and $18^{th}$ tasks ($0^{th}-4^{th}$, $5^{th}-9^{th}$, $45^{th}-49^{th}$, and $90^{th}-94^{th}$ train shots), respectively.

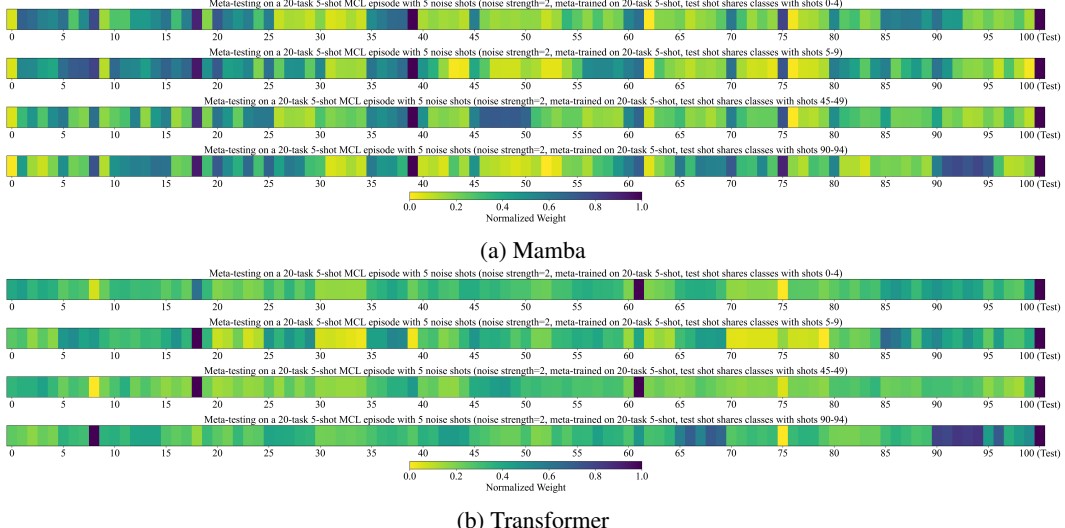

(a) Mamba

(b) Transformer

Figure 14: Noise inputs in meta-testing: visualization of the final layer associations between various test shots (queries) and a single MCL train episode (prompt) of both (a) Mamba and (b) Transformer (meta-trained on 20-task 5-shot without noise inputs), during meta-testing on 20-task 5-shot MCL episode with noise inputs (noise strength=2, noise on $8^{th}$, $18^{st}$, $39^{th}$, $61^{th}$, and $75^{th}$ training shots). In meta-testing, the four visualizations share a single MCL training episode (prompt) spanning $0^{th}$–$99^{th}$ shots, while the test shots (queries at the $100^{th}$ shot) correspond to the $0^{th}$, $1^{st}$, $9^{th}$, and $18^{th}$ tasks ($0^{th}-4^{th}$, $5^{th}-9^{th}$, $45^{th}-49^{th}$, and $90^{th}-94^{th}$ train shots), respectively.

Mamba performs more effectively by selectively associating relevant information and leveraging its recurrently updated latent state, which accumulates global sequence information.

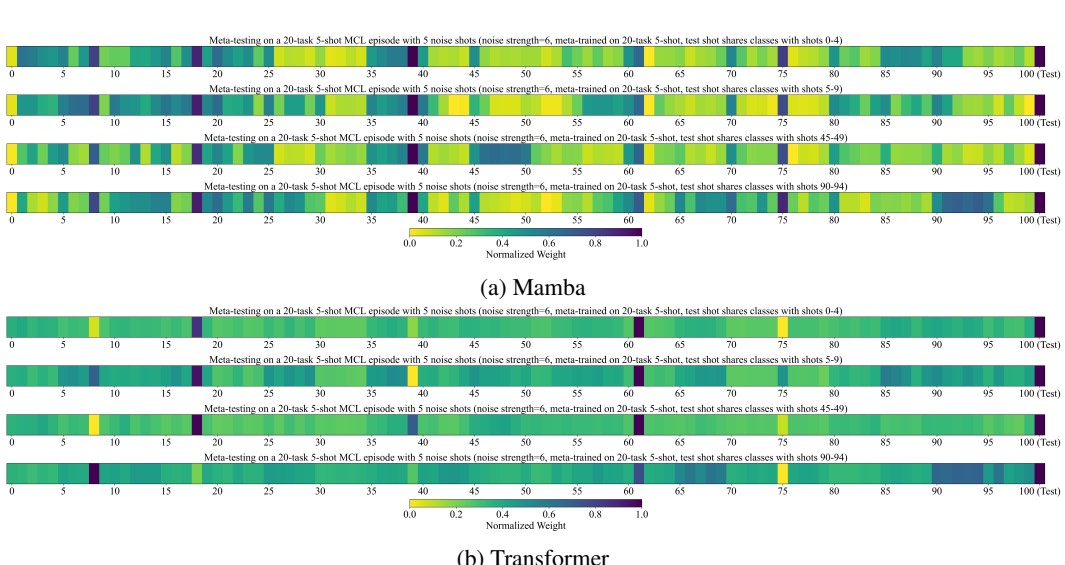

Figure 15: Noise inputs in meta-testing: visualization of the final layer associations between various test shots (queries) and a single MCL train episode (prompt) of both (a) Mamba and (b) Transformer (meta-trained on 20-task 5-shot without noise inputs), during meta-testing on 20-task 5-shot MCL episode with noise inputs (noise strength=6, noise on $8^{th}$, $18^{st}$, $39^{th}$, $61^{th}$, and $75^{th}$ training shots). In meta-testing, the four visualizations share a single MCL training episode (prompt) spanning $0^{th}$– $99^{th}$ shots, while the test shots (queries at the $100^{th}$ shot) correspond to the $0^{th}$, $1^{st}$, $9^{th}$, and $18^{th}$ tasks ($0^{th}-4^{th}$, $5^{th}-9^{th}$, $45^{th}-49^{th}$, and $90^{th}-94^{th}$ train shots), respectively.

