# OpenReview forum: "Learning Mamba as a Continual Learner"
_ICLR.cc/2025/Conference — Submitted to ICLR 2025_

### Official Review · Reviewer_BA8y · 2024-10-27

**Soundness:** 2
**Presentation:** 2
**Contribution:** 2
**Rating:** 5
**Confidence:** 4

**Summary:**

The authors explore a key research question: Can the attention-free Mamba model effectively handle meta-continual learning (MCL) tasks? They reframe State Space Models (SSM) and Mamba as sequence-prediction-based continual learners, training them via meta-learning across continual learning episodes. To enhance this training, they introduce a selectivity regularization technique. Extensive experiments reveal that Mamba consistently performs well in various MCL settings, significantly surpassing other attention-free approaches and often equaling or surpassing Transformer models in performance—all while using fewer parameters and computational resources. Notably, Mamba demonstrates strong reliability, generalization, and robustness in complex scenarios.

**Strengths:**

* It is interesting to explore how  Mamba performs in a meta-continual learning setting.

**Weaknesses:**

* The conclusion of this paper is unsurprising, as Mamba's MCL performance aligns closely with its results on standard benchmarks.

* There is insufficient analysis explaining how and why Mamba outperforms other attention-free architectures and achieves comparable results to Transformer-based models.

**Questions:**

N/A

---

> ### Author Response · Authors · 2024-11-23
> **Response by authors**
>
> Thank you very much for your time and effort in reviewing our paper.
> >**W1. The conclusion of this paper is unsurprising, as Mamba's MCL performance aligns closely with its results on standard benchmarks.**
>
> Thank you for your comments.
> - We would like to kindly discuss the notable results presented in our manuscript.
> Unlike (Lee et al., 2024), where attention-free methods fail to match the performance of Transformers, our work demonstrates the successful development of an attention-free model for MCL. It's not strightforward or trivial to apply the Mamba model to MCL. Considering the unique characteristics of Mamba, we developed a selectivity regularization technique and applied Mamba effectively in MCL for the first time.
> Additionally, we conduct extensive investigations and analyses of model performance across more diverse and complex scenarios, providing further insights.
> - Transformers resemble RNNs superficially but are fundamentally different because they require recomputing the full attention map at each step with a complete KV-cache, which contradicts the principles of continual learning to some extent.
> Our results demonstrate improved efficiency and comparable or better performance in specific scenarios compared to vanilla Transformers (with full KV-cache), underscoring the potential of attention-free models in MCL.
> - The observations and conclusions from our analyses and experiments align with prior studies on Mamba in other applications (e.g., language modeling), supporting the validity of our results. Achieving these findings in the context of MCL is both non-trivial and novel.
>
> ---
>
> >**W2. More analysis explaining how and why Mamba outperforms other attention-free architectures and achieves comparable results to Transformer-based models?**
>
> Thank you for your comments. We have incorporated additional analysis to explain how and why Mamba outperforms better in Appendix C and D.
>
> - Mamba performs better generalization to untrained stream length.
>     - The experiments shown in Figures 3a and 3b validate the generalization ability of models by meta-testing on CL episodes/sequences that differ from those seen during meta-training. Transformers generally converge more easily during meta-training compared to Mamba, due to their strong fitting ability. However, this advantage may also lead to meta-overfitting.
>     - To analyze how different models perform on these sequences, we visualize the final-layer attention weights of Transformers and the corresponding selective scores (associative indicators) of Mamba. Note that Mamba does not have explicit attention weights, we the scores relying on the connection between Mamba and Transformers described in Section 3.2.2. For models meta-trained on the 20-task, 5-shot setting, we meta-tested them and visualized their weights on 20-task, 5-shot episodes (Fig. 10), 20-task, 10-shot episodes (Fig. 11), and 40-task, 5-shot episodes (Fig. 12).
> 	- Specifically, we observed that Transformers tend to either average attention or consistently focus on specific token positions in episodes that deviate from the training length. In contrast, Mamba effectively associates with relevant shots. This suggests that Transformers may learn pattern biases in the sequences (e.g., positional biases unrelated to content), leading to meta-overfitting during these generalization tests.
>
> -   Mamba performs better robustness to large input noise.
>     - We visualized the final layer attention weights for test shots compared to training shots for both Mamba and Transformer, each meta-trained in a 20-task, 5-shot setting. During meta-testing, these models processed a 20-task, 5-shot episode with five noisy input shots (shot index: 8, 18, 39, 61, 75) at noise strengths of 1 (Fig. 13), 2 (Fig. 14), and 6 (Fig. 15).
>     - The results indicate that Transformers meta-trained on clean episodes tend to produce extreme attention weights (either very high or very low) on noisy or outlier shots, whereas Mamba is less affected. This observation suggests that Transformers’ learned attention mechanisms tend to associate samples based on local and independent representations. In contrast, Mamba performs more effectively by selectively associating relevant information and leveraging its recurrently updated latent state, which accumulates global sequence information.

---

> ### Author Response · Authors · 2024-12-02
> **Thank you for your comments**
>
> Thank you very much for acknowledging the notable strengths of MambaCL, including its reliability, generalization, and robustness in complex scenarios. We also appreciate your recognition of the novelty of exploring Mamba's performance in MCL setting.
>
> We deeply appreciate your time and effort in reviewing our work. We hope that our responses have been satisfactory and welcome any further discussion. Should our rebuttal sufficiently address your comments, we kindly request that you consider increasing the score. Thank you for your valuable feedback.

---

> > ### Comment · Reviewer_BA8y · 2024-12-02
> >
> > Thanks for the author's rebuttal. After reading the comments from other reviewers, the reviewer thinks that this paper needs to be further improved. I maintain my score.

---

### Official Review · Reviewer_bJ6d · 2024-11-02

**Soundness:** 3
**Presentation:** 3
**Contribution:** 2
**Rating:** 6
**Confidence:** 4

**Summary:**

This work addresses meta-continual learning using a state space model Mamba. It performs comprehensive experiments across various CL benchmarks and reports several interesting results, including comparison with Transformers and extension to Mamba mixture-of-experts.

**Strengths:**

1. It proposes MambaCL as a strong sequential approach to meta-continual learning.

2. It performs thorough experiments and discover multiple interesting observations.
- The use of Mamba may be more helpful for generalization over Transformers as discussed in Fig.3.
- MambaCL is particularly effective in on fine-grained recognition tasks as shown in Table 3.
- Integration of Mamba with MoE improves the MCL performance as reported in Table 6.

**Weaknesses:**

1. The technical novelty is limited.
- This work is largely based on the work of (Lee et al., 2024), which first formulates the MCL problem as a sequent modeling.
- This work simply replaces Transformers of (Lee et al., 2024) with a state space model Mamba.
- Except this replacement, there is little novelty as its application is rather straightforward, following (Lee et al., 2024).

2. The use of Mamba instead of Transformers leads to little performance improvement as reported in Table 1-5.
- The main benefit of Mamba over Transformer lies in fewer parameters and increased processing speed as shown in Table 7.

3. Implementation details are missing.
- Appendix is too sketchy to fully understand how the MambaCL is implemented.
- The code is not provided.

**Questions:**

Please see the Weaknesses.

---

> ### Author Response · Authors · 2024-11-23
> **Response by authors**
>
> Thank you very much for your time and effort in reviewing our paper.
> >**W1. Novelty and relationship with (Lee et al., 2024).**
>
> > - This work is largely based on the work of (Lee et al., 2024), which first formulates the MCL problem as a sequent modeling.
> > - This work simply replaces Transformers of (Lee et al., 2024) with a state space model Mamba.
> > - Except this replacement, there is little novelty as its application is rather straightforward, following (Lee et al., 2024).
>
> Thank you for your comments.
> - We want to emphasize the novelty of our work across three key aspects, i.e., problem formulation, analytical insights on methods, and technical contributions.
>     - Our work shares a similar meta-continual learning formulation as (Lee et al., 2024). However, we emphasize that this formulation represents a general problem deserving further investigation. Beyond the basic formulation and tasks explored in (Lee et al., 2024), our work extends these investigations to broader and more realistic scenarios, such as generalization to diverse meta-test tasks and robustness to noisy scenarios. These are novel perspectives and distinguish our approach from (Lee et al., 2024).
>     - For the methodoloy aspect, our work identifies a key gap between attention-based Transformers (which store K and V for all seen samples) and suitable methods for meta-continual learning (MCL). To address this, we focus on the studies of attention-free MCL models, which sets our work apart from (Lee et al., 2024) and provides a novel direction for MCL research. Considering the significant potential of MCL, our work expands its applicability to more realistic scenarios, making substantial contributions to the field.
>     - On technical aspect, we introduce the attention-free Mamba model tailored to the MCL formulation and propose specific techniques to ensure its effectiveness. This represents a novel contribution. Unlike (Lee et al., 2024), where attention-free methods fail to match the performance of Transformers, our work demonstrates the successful development of an attention-free model for MCL. It is non-trivial. Additionally, we conduct extensive investigations and analyses of model performance across more diverse and complex scenarios, providing further insights.
>
>
> ---
> >**W2. The use of Mamba instead of Transformers leads to little performance improvement as reported in Table 1-5. The main benefit of Mamba over Transformer lies in fewer parameters and increased processing speed as shown in Table 7.**
>
> Thank you for your comments.
> - Firstly, we want to highlight that our motivation for investigating attention-free models (e.g., Mamba, as presented in our paper) for meta-continual learning (MCL) is rooted in its alignment with the principles of continual learning. Although Transformers have demonstrated strong performance (Lee et al., 2024), relying solely on Transformers limits the broader applicability of MCL.
>     - Unlike Transformers, which store K and V for all seen samples and require a linearly increasing hidden state size, attention-free models maintain a constant hidden state size. This aligns better with the requirements and definitions of continual learning (CL), where efficiency and scalability are critical. Our primary motivation lies in addressing these natural characteristics of CL, rather than focusing solely on performance.
>
>
> - Mamba is an attention-free model. Transformers resemble RNNs superficially but are fundamentally different because they require recomputing the full attention map at each step with a complete KV-cache, which contradicts the principles of continual learning to some extent.
> - We conducted comprehensive experiments to evaluate the effectiveness of various methods. We observed that Mamba outperforms other attention-free methods by leveraging time-variance selective modeling.
> - While our study does not specifically aim to establish Mamba’s superiority over Transformers, Mamba, with its significantly smaller state size and greater efficiency, can avhieve performance comparable to or even exceeding that of attention-based Transformers, particularly in scenarios requiring long-term structure modeling, as detailed in the paper.
>
> ---
>
> >**W3. Implementation details.**
>
> Thank you for your comments. We have further emphasized and expanded the implementation details in both the following responses and the revised manuscript. We also commit to releasing all code upon the acceptance of the manuscript.
> - The experimental setup and implementation details are provided in Chapter 4 (Lines 332–357). Fig. 1 illustrates the overall MCL process, while Fig. 2 focuses on the detailed Mamba block, designed following the standard Mamba structure. Additionally, Appendix B includes configurations of the various models used in our experiments.
> - If there are any further questions or points that need clarification, please let us know, and we will do our best to address them.

---

> ### Author Response · Authors · 2024-12-02
> **Thank you for your comments**
>
> Thank you very much for recognizing the interesting results and the strengths of Mamba in MCL, including its generalization capabilities and robustness across various benchmarks. We appreciate your acknowledgment of the proposal of MambaCL as a strong sequential approach and the extension to MambaCL-MoE.
>
> We deeply appreciate your time and effort in reviewing our work. We hope that our responses have been satisfactory and welcome any further discussion. Should our rebuttal sufficiently address your comments, we kindly request that you consider increasing the score. Thank you for your valuable feedback.

---

### Official Review · Reviewer_Zeh1 · 2024-11-09

**Soundness:** 2
**Presentation:** 3
**Contribution:** 2
**Rating:** 3
**Confidence:** 4

**Summary:**

The paper follows the meta continual learning (MCL) framework as outlined by Lee et al., 2024. The authors meta-train sequential models on offline meta-training sequences to enhance their sequence modelling capability. The authors propose using Mamba as the sequential model instead of transformers or attention-free variants to alleviate high computational costs while still achieving satisfactory performance. Additionally, the authors introduce a selective regularization technique for meta-training, which enhances the association between query tokens and previously correlated input tokens. Experimental results demonstrate that Mamba achieves improved generalization and robustness compared to transformer variants in MCL, while using less memory for inference.

**Strengths:**

- The paper is well-structured and easy to follow.
- The authors clearly explained the issue of increased compute complexity with using transformers for MCL.

**Weaknesses:**

In general:

- The paper shows limited novelty. The problem formulation, specifically the recasting of the continual learning problem as a sequential modelling problem in recurrent models, mirrors the previous work by Lee et al., 2024. From the technical side, the authors propose a new selective regularization technique for meta-training and claim it improves training stability and convergence. While the technique itself is novel, there are several questionable aspects regarding this technique and the authors' claims. I cannot fully credit the novelty of this technique until these issues are addressed.

- Although the authors claim better generalization and robustness when using Mamba instead of transformers based on empirical results, these results appear somewhat questionable. Furthermore, there is a lack of new insights and detailed analysis; for instance, the authors did not delve deeper into the underlying mechanisms that led to these results. This deeper analysis is crucial, especially if the primary motivation of the paper is to use Mamba (or any different model architecture) instead of transformers for the same problem settings.

Please kindly refer to the questions for more details.

**Questions:**

I am open to discussion and willing to reconsider my score if my major concerns can be adequately addressed.


**Claims on the Effectiveness of the Proposed Regularization Technique**

- For example, lines 326-329 state:
  > We apply this regularization to MambaCL and other sequence prediction models (weighted by a scalar λ) together with the MCL objective in Eq. (7), which improves the meta-training stability and convergence for all models.

- The authors do not fully support their claims about "improving the meta-training stability and convergence for all models." Specifically, there are no experiments showing learning curves (or similar alternatives) for all models during meta-training to compare results with and without this technique.

- A seemingly related empirical evidence is presented in Figure 4. However, the results appear to pertain to a *single* model, and it is unclear, based on the figure caption and the text in lines 481-485, which specific model (i.e., Mamba, transformers) was used in this ablation study. Although the experiment demonstrates the sensitivity of meta-testing performance to the regularization strength, it lacks comprehensive evidence across multiple models to support the authors claim.


**Experiment Implementation Details**

- In the paper, it is mentioned:
  > Following Lee et al., 2024, we set the initial learning rate to 1 × 10⁻⁴...

- Cloud the authors please provide some motivations for using the same hyperparameters as in Lee et al., 2024, given that the meta-training setups differ? Specifically, the authors used a pre-trained CLIP backbone as a visual encoder and included the proposed regularization loss across all models.

- Moreover, were these hyperparameters adjusted for different model architectures based on some meta-validation sets, e.g., for linear transformers and Mamba? If not, wouldn't using fixed hyperparameters for all experiments and models potentially lead to sub-optimal results? If these hyperparameters are not optimal for every models, this could produce misleading results and potentially invalidate the observations.

**Meta-Overfitting in Figures 3a and 3b**

- The authors observed that transformers and their variants seem to suffer from severe meta-overfitting based on the results in Figures 3a and 3b. However, the potential underlying causes for this overfitting are quite unclear. Specifically:

  - As previously mentioned, based on the current description of the implementation details, it's unclear whether this overfitting is due to the use of improper hyperparameters, such as learning rates.

  - Additionally, it is undetermined whether this overfitting is influenced by the use of regularization terms for all models during meta-training. Would removing this regularization loss for transformers significantly reduce meta-overfitting?

- Could the authors please provide some insights into why Mamba did not suffer from the same degree of overfitting?

- While the occurrence of meta-overfitting is expected, the degree of overfitting—particularly in relation to the number of training tasks and training shots used in meta-training—exhibited by transformers and their variants in Figures 3a and 3b is somewhat surprising. Specifically, in Figure 3b, adding more training shots per class even, and almost monotonically, decreased the classification accuracy on the queries.


**Robustness in Figure 3c**

- It is somewhat unclear how the authors performed the input noise perturbation. Specifically, what does $ x_i$ in line 473 refer to? Is it the original input image to the CLIP encoder, or the extracted image embeddings that serve as inputs to the sequential learning models?

- I find it very interesting that Mamba exhibits excellent robustness to input noise, even with a standard deviation as large as 10. Could the authors potentially discuss some potential reasons behind Mamba's extreme robustness to large input noise?

**General Comments on MCL**

- Some important challenges in the MCL setup for continual learning include: 1) its application to long continual learning sequences, 2) the requirement for offline training datasets (meta-training), and 3) generalization to unseen long OOD meta-testing tasks. These challenges cannot be resolved simply by switching from transformers or their variants to Mamba.

- Are there any differences on the problem formulation and the meta-training setups between the ones in the paper and the one in MetaICL: Learning to Learn In Context, Min et al., NAACL 2022?

---

> ### Author Response · Authors · 2024-11-23
> **Response by authors (Part 1)**
>
> Thank you very much for your time and effort in reviewing our paper.
> >**W1. Novelty. Formulation relationship with (Lee et al., 2024). Technical novelty.**
>
> Thank you very much for your comprehensive comments.
>
> We want to emphasize the novelty of our work across three key aspects, i.e., problem formulation, analytical insights on methods, and technical contributions.
>
> - Our work shares a similar meta-continual learning formulation as (Lee et al., 2024). However, we emphasize that this formulation represents a general problem deserving further investigation. Beyond the basic formulation and tasks explored in (Lee et al., 2024), our work extends these investigations to broader and more realistic scenarios, such as generalization to diverse meta-test tasks and robustness to noisy scenarios. These are novel perspectives and distinguish our approach from (Lee et al., 2024).
>
> - For the methodoloy aspect, our work identifies a key gap between attention-based Transformers (which store K and V for all seen samples) and suitable methods for meta-continual learning (MCL). To address this, we focus on the studies of attention-free MCL models, which sets our work apart from (Lee et al., 2024) and provides a novel direction for MCL research. Considering the significant potential of MCL, our work expands its applicability to more realistic scenarios, making substantial contributions to the field.
>
> - On technical aspect, we introduce the attention-free Mamba model tailored to the MCL formulation and propose specific techniques to ensure its effectiveness. This represents a novel contribution. Unlike (Lee et al., 2024), where attention-free methods fail to match the performance of Transformers, our work demonstrates the successful development of an attention-free model for MCL. It is non-trivial. Additionally, we conduct extensive investigations and analyses of model performance across more diverse and complex scenarios, providing further insights.
>
> We will address the specific concerns raised in the following points, ensuring clarity any confusion regarding our proposed techniques.
>
> ---
>
>
> >**W2. Results of Mamba comparing to Transformer; Mechanisms leading to the results; "... deeper analysis is crucial, especially if the primary motivation ... is to use Mamba ... instead of transformers for the same problem settings."**
>
> Thank you for your comments.
>
> - The motivation for investigating attention-free models (e.g., Mamba, as presented in our paper) for meta-continual learning (MCL) stems from their intrinsic advantages. Unlike Transformers, which store K and V for all seen samples and require a linearly increasing hidden state size, attention-free models maintain a constant hidden state size. This aligns better with the requirements and definitions of continual learning (CL), where efficiency and scalability are critical. Our primary motivation lies in addressing these natural characteristics of CL, rather than focusing solely on performance.
>
>
> - Given that all attention-free methods studied in (Lee et al., 2024) fail to match the performance of Transformers, we developed an advanced Mamba model tailored for MCL to explore the potential. All experiments involving different models were conducted under the same conditions, ensuring fairness and consistency with prior methods. Mamba demonstrates superior performance compared to other attention-free methods, leveraging time-variance selective modeling. This enables it to align with or even surpass the performance of attention-based Transformers, particularly in scenarios that depend on long-term structures, as discussed in the paper. While the studies and observations in our work are novel, the results are consistent with prior studies on Mamba in other applications (e.g., language modeling).
>
> Further analyses and insights are demonstrated in the following to address the suggested clarifications.

---

> > ### Author Response · Authors · 2024-11-29
> > **Thank you for your comments (Round 2 Part 2/3)**
> >
> > > **Q3. ...I am aware one technical change being made is the regularization proposed in this paper. While the regularization technique does contribute to stabilizing the meta-training process, I am concerned that the observed results (regarding better generalization of Mamba) may largely stem from the use of the Mamba model itself rather than this algorithmic development...**
> >
> > 1) Firstly, in principle, directly using any existing Mamba model and implementation for other tasks cannot work on our task, i.e., MCL. We can see many applications of the Mamba model (and Transformers) on different applications, which are all Mamba models (or Transformers, such as (Lee et al., 2024)). Although new techniques are not our sole contributions, considering that the reviewer has recognized and admitted our main novel techniques and contributions and we further highlighted more detailed technical novelties, we think it has been clear that our work is clearly different to existing works.
> > 2) > “This technique appears to have very limited effectiveness on the attention-free model beyond Mamba, which still resulted in significant meta-overfitting.”
> >
> >     - Firstly, we never claimed to use this regularization loss to solve the “meta-overfitting” issue. We think the reviewer has a misunderstanding here.
> >     - Without such a regularizer, meta-training of all models is difficult and almost cannot converge to a satisfactory solution (as discussed in Sec. C.1). The regularizer is used to lead the training converge. Our main contribution is proposing this regularizer to Mamba by bridging the selective operation and attention. And we apply similar loss for all models in experiments for fair comparison. Such regularization is also effective for other models for convergence, which is also observed in (Lee). In our rebuttal, we only want to highlight that the “meta-overfitting” (mentioned by the reviewer) is not related to this regularization loss.
> >     - The “meta-overfitting” behavior of Transformers in Fig. 3 may be mainly related to the model design. This observation can align with the analysis of Transformer and Mamba (Park et al., 2024; Garg et al., 2022; Bornschein et al., 2024).
> >
> >
> >
> >
> > 3) > “The authors mentioned that generalization performance is relatively insensitive to the strengths of the regularization (Fig. 9 and 3). This raises questions about the role of the regularization technique in achieving the reported improvements for Mamba.”
> >
> >     - Without this regularization, the models cannot converge stably and cannot converge to satisfactory results. We show this in Fig. 6.
> >     - Fig. 8 shows that the models are not sensitive to the setting of the hyperparameters in a proper and large range, which is a good property of the technique. The experiments are conducted in a standard way. If we set the hyperparameter as a very small or very large value, the performance will be changed.
> >
> >
> > 4) > “As a result, would you agree that the empirical performance improvements observed in comparison to Transformers — specifically in memory efficiency and generalization to longer input sequences — are more likely inherent properties of the Mamba model itself, which were already highlighted in the original Mamba paper?”
> >
> >     - The good performance of the Mamba model is from the design of the Mamba model. It is also what we want to highlight in the paper: the attention-free model, like Mamba, can also perform well on MCL. It is different from the previous work (Lee et al., 2024).
> >     - We do not agree that they `“were already highlighted in the original Mamba paper”`. The results on language modeling in the original paper cannot be directly applied or extended to other domains. That is also why it has been important to investigate how to generalize different models like Mamba (Park et al., 2024) and Transformers (Garg et al., 2022; Bornschein et al., 2024) to other tasks.
> >     - For MCL, (Lee et al., 2024) shows that the attention-free methods cannot perform well with a significant gap to the Transformer, although attention-free methods like Linear Transformer have been proven effective on language tasks. We provide novel insights, new techniques, and empirical results to make the attention-free method (e.g., Mamba) work on MCL. As we highlighted several times, making Mamba work well on MCL is not trivial where the essential component is the associative regularization and other details.

---

> ### Author Response · Authors · 2024-11-23
> **Response by authors (Part 2)**
>
> >**Q1. Effectiveness of the Proposed Regularization Technique**
>
> - > "improving the meta-training stability and convergence for all models."; learning curves for all models during meta-training with and without selectivity regularization technique.
>     - Due to the complexity of the MCL task, the proposed regularization technique plays a crucial role in stabilizing and improving the training process for all models. To highlight its impact, we have added meta-training loss curves in Fig. 6 of the revised manuscript, showing the initial 2500 steps for different models with and without selectivity regularization. The results indicate that models without our regularization struggle to converge and exhibit significant oscillations during training, highlighting the effectiveness of the regularization.
>
> - > Regularization strength ablation study across multiple models.
>     - Applying regularization to Transformers or Linear Transformers is straightforward, as it involves direct attention map adjustments. However, for Mamba’s selective modeling, which implicitly bridges it with Transformer architectures, this process is more complex. Thus, the ablation study in Fig. 4 focuses primarily on the Mamba model.
>     - To address the reviewer’s concern, we have expanded the revised manuscript to include the sensitive study of the regularization strength ($\lambda$) for different models, as shown in Appendix C.2 (Fig. 7). The results demonstrate that all models exhibit stability within a wide and appropriate range of $\lambda$, providing evidence of consistent patterns.
>
>
> ---
> > **Q2. Experiment Implementation Details**
>
>  - >Why using same hyperparameters?
>
>     Thank you for pointing out this.
>
>     - During the initial phase of this project, we started by experimenting using the same learning rate hyperparameter as in previous works. Through experimentation with different learning rates, we observed that both our model and others (under the same setting, tasks, and input types) were largely insensitive to this parameter across a wide range. To ensure a fair comparison, we therefore adopted the same learning rate as used in (Lee et al., 2024) for the compared methods, by default. We acknowledge that the term "following" is confusing and have corrected this in the revision. We have added more implementation details in the revised version of the manuscript. We commit to releasing all code upon the acceptance of the manuscript.
>     - Furthermore, we have included Fig. 8 in Appendix C.3 that illustrates the performance of the models using various initial learning rates $\{5\times10^{-5}, 1\times10^{-4}, 2\times10^{-4}, 5\times10^{-4}\}$ on both ImageNet-1K and CIFAR-100 datasets in Fig. 8. The results indicate that within a reasonable range, the learning rate does not significantly affect model performance. In our experiments, we set the initial learning rate to $1\times10^{-4}$, with decays of 0.5 every 10,000 steps.
>
>  - > Are the hyperparameters adjusted?
>      - As discussed above, we observed that the models’ behavior was largely insensitive to these hyperparameters. Consequently, we did not perform extensive optimization or search for optimal hyperparameter settings. Instead, we adhered to the experimental settings outlined by Lee et al. (2024) by default. As demonstrated in the hyperparameter sensitivity analyses (e.g., $\lambda$ in Fig. 7 and learning rate in Fig. 8 the chosen hyperparameter settings do not affect the results. We acknowledge that the hyperparameter settings also do not influence the conclusions.

---

> > ### Author Response · Authors · 2024-11-29
> > **Thank you for your comments (Round 2 Part 3/3)**
> >
> > > **Q4. More on this regularization technique. I agree that it stabilizes meta-training, but I am still not sure if I understood the rationale behind applying this technique to all models, as this would prevent us from seeing how different models behave intrinsically.
> > Initially, I thought meta-training was impossible without this technique, but it seems that (Lee et.al, 2024) managed to produce meaningful results without this technique. Although the meta-training losses, in the new Figures, showed more oscillation, they still showed a clear decreasing trend indicating convergence...**
> >
> > 1) If without such kind of regularizations, all models cannot  converge to a reasonable solution. Fig. 6 showing the initial training phases (2500 steps) for different models with and without selectivity regularization. The losses are 3–5 times higher compared to the models with regularization applied and successfully converging. Beyond 2500 steps, the losses oscillate and no longer decrease. `“meta-training was impossible without this technique”` – it is correct.
> > 2)  - Note that the reported results of Transformers and Linear Transformers in the paper of (Lee et al., 2024) also rely on such kind of regularization. The Transformer implementation of (Lee et al., 2024) cannot produce reasonable results without such kind of regularization.
> >      - It is straightforward to regularize the attention map of Transformers. But there is no explicit attention or association process in SSM-based Mamba, our novelty is mainly on proposing the regularization for our Mamba model relying on bridging Mamba/SSM and Transformer.
> >      - We do not want to highlight the effectiveness of such regularization on other models. We mentioned the performance of the regularization on other models to address your concerns related to the effects or this regularization on others. And we only highlight that such kind of regularization is used for all models for fair comparison in implementation. The main information to deliver is only that (a) the regularization helps the meta-training of different models (consistent with the discussions of (Lee et al., 2024)), (b) the performance issues of Transformers (such as “meta-overfitting”) are not caused by this regularization. We apologize for the potential confusion may be caused by the wording in the rebuttal.
> >
> >
> >
> > ---
> > > **Q5. I have some reservations about the work provides a novel direction for MCL in the use of attention-free methods. As previously mentioned, the improvements attributed to the empirical improvements over transformers seem to be closely tied to the specific characteristics of the Mamba model. The significant meta-overfitting observed in other attention-free methods suggests that the broader applicability of these models for MCL may be limited by the specific configurations of your meta-training setup.**
> >
> > We want to clarify and highlight the motivation and the rationale again. We focus on attention-free models for MCL not only for their efficiency or effectiveness on performance.
> > 1) MCL is studied to meta-learn continual learners for CL without the need to maintain all previously seen methods.
> > 2) Transformer needs to maintain the K and V of all seen samples. Despite good numerical performances reported in (Lee et al., 2024), Transformer is not an ideal or proper choice for MCL, due to the misalignment with the objective of MCL.
> > 3) We highlight this issue of the Transformer and propose to focus on attention-free methods, which do not need to save representations of all samples. We focus on the direction of using attention-free models for MCL because their design aligns with the definition of MCL better.
> > 4) Although attention-free models fit the objective of MCL better, all attention-free models cannot work well as reported by (Lee et al., 2024) (and also as you can recognize). Note that other attention-free models can perform well on many general language-based applications, which is inconsistent with the unsatisfactory performances on MCL. (It can also show that the effectiveness of a model on language-based applications does not naturally mean effectiveness on MCL.) We thus focus on studies of model Mamba on MCL and show that an attention-free model can also perform well on MCL in more scenarios. This observation is novel compared to previous work (Lee et al., 2024).

---

> ### Author Response · Authors · 2024-11-23
> **Response by authors (Part 3)**
>
> > **Q.3 Meta-Overfitting illustrated in Figures 3a and 3b**
>
>
> - > More insights for why Mamba performs better on this experiment.
>     - The experiments shown in Figures 3a and 3b validate the generalization ability of models by meta-testing on CL episodes/sequences that differ from those seen during meta-training. Specifically, the models are meta-trained on “20-task, 5-shot” MCL episodes and meta-tested on episodes with task and shot numbers exceeding those in meta-training. It is also a novel perspective we want to investigate in the paper.
>     - Transformers generally converge more easily during meta-training compared to Mamba, due to their strong fitting ability. However, this advantage may also lead to meta-overfitting.
>     - To analyze how different models perform on these sequences, we visualize the final-layer attention weights of Transformers and the corresponding selective scores (associative indicators) of Mamba. Note that Mamba does not have explicit attention weights, we the scores relying on the connection between Mamba and Transformers described in Section 3.2.2. For models meta-trained on the 20-task, 5-shot setting, we meta-tested them and visualized their weights on 20-task, 5-shot episodes (Fig. 10), 20-task, 10-shot episodes (Fig. 11), and 40-task, 5-shot episodes (Fig. 12).
> 	- Specifically, we observed that Transformers tend to either average attention or consistently focus on specific token positions in episodes that deviate from the training length. In contrast, Mamba effectively associates with relevant shots. This suggests that Transformers may learn pattern biases in the sequences (e.g., positional biases unrelated to content), leading to meta-overfitting during these generalization tests.
>
>
> - > Does the setting of hyperparameters (learning rate) affect meta-overfitting?
>     - As discussed above, Fig. 8 illustrates that the models maintain robustness across a reasonable range of learning rates. In practice, we observe that models with different learning rates encounter similar behaviours.
> - > Does selectivity regularization affect meta-overfitting?
>     - Without the regularization, models struggle to converge and exhibit significant oscillations during training, as shown in Fig. 6. Therefore, we evaluated various models with a small regularization strength (0.1) to assess the impact of regularization on this generalization experiment and the meta-overfitting issue. The results indicate that regularization strengths of 0.1 (Fig. 9) and 0.5 (Fig. 3) lead to similar phenomena across different models. This experiment and the $\lambda$ sensitivity analysis (Fig. 7) show that the results are not influenced by the hyperparameter setting.
>
> - > "... transformers and their variants in Figures 3a and 3b is somewhat surprising. Specifically, in Figure 3b, adding more training shots per class even, and almost monotonically, decreased the classification accuracy on the queries."
>     - The continual learning ability of the MCL models is given by the meta-training. Although more samples an episode/sequence contain more information, the models can perform significantly worse given the episodes different to those seen in meta-training. The models in this exepriment (e.g., Transformers) are trained using a 5-shot pattern, and providing additional shots might lead the model to mistakenly perceive them as belonging to different tasks potentially leading to overfitting (given that Transformers may learn to associate every 5 shots with each task indicated by the position encoding).

---

> ### Author Response · Authors · 2024-11-23
> **Response by authors (Part 4)**
>
> > **Q.4 Robustness of Noise Input in Figure 3c**
>
> - > How the noise is added.
>     - Thank you for pointing out this. The noise is added on the input of the model, i.e., $x_i$. In this context, $x_i$ represents the image embeddings extracted from the pre-trained CLIP model. We have revised the manuscript to clarify it, accordingly.
>
> -   > Could the authors potentially discuss some potential reasons behind Mamba's extreme robustness to large input noise?
>     - In the experiments, the modes are meta-trained on noise-free episodes. And the noise is added on randomly selected samples/shots in the meta-testing episodes. The task can also be seen as validating the ability of ignoring the irrelevant samples or contaminated outlier samples in the sequences.
>
> - To directly show how the models work in this scenarios, we visualized the final layer attention weights for test shots compared to training shots for both Mamba and Transformer, each meta-trained in a 20-task, 5-shot setting. During meta-testing, these models processed a 20-task, 5-shot episode with five noisy input shots (shot index: 8, 18, 39, 61, 75) at noise strengths of 1 (Fig. 13), 2 (Fig. 14), and 6 (Fig. 15).
>     - The results indicate that Transformers meta-trained on clean episodes tend to produce extreme attention weights (either very high or very low) on noisy or outlier shots, whereas Mamba is less affected. This observation suggests that Transformers’ learned attention mechanisms tend to associate samples based on local and independent representations. In contrast, Mamba performs more effectively by selectively associating relevant information and leveraging its recurrently updated latent state, which accumulates global sequence information.
>
>
> ---
> > **Q5. General comments on MCL**
>
> - > Some important challenges in the MCL setup for continual learning include: 1) its application to long continual learning sequences, 2) the requirement for offline training datasets (meta-training), and 3) generalization to unseen long OOD meta-testing tasks. These challenges cannot be resolved simply by switching from transformers or their variants to Mamba.
>     - Thank you for the comments. The pointed challenges are also our motivation and the novel perspectives explored in our paper, distinguishing our work from (Lee et al., 2024). We aim to address these challenges from the broader perspective of extending MCL to more realistic and practical scenarios.
> 	- In this study, our goal is not to resolve these challenges using a single specific model, such as Mamba.
> 	- Our motivation for studying Mamba in the context of MCL is rooted in its alignment with the principles of continual learning. Unlike attention-based Transformers, an attention-free model (e.g., Linear Transformer or Mamba) does not require maintaining representations for all seen samples, making it inherently more suitable for continual learning.
>
>
> - > Discussion of differences with MetaICL
>     - Thank you for pointing this out. We have included a discussion in the revision. MetaICL is designed for language text, whereas our tokens include images and labels. The underlying functions to be fitted ('the functions to fit') are distinct, although they share a common mathematical formulation. Compared to text sequences, the problems we address are inherently more complex, requiring the learning of more intricate functions and making the learning process more challenging.

---

> > ### Comment · Reviewer_Zeh1 · 2024-11-28
> > **Thank you for your detailed response (part 1/2)**
> >
> > Dear Authors,
> >
> > I apologize for the delayed response.
> >
> > I sincerely appreciate the time and effort the authors have dedicated to providing a detailed response to my review and making revisions to the original manuscript. I would like to highlight a few positive aspects:
> > - Thank you for providing additional details regarding noise perturbation, hyperparameters, and experimental settings.
> > - I recognize the effectiveness of the proposed regularization technique in stabilizing meta-training.
> > - I appreciate the additional visualizations of attention scores for the models.

---

> ### Comment · Reviewer_Zeh1 · 2024-11-28
> **Thank you for your detailed response (part 2/2)**
>
> I genuinely value your participation in the rebuttal and discussion period. However, at the current stage, some of my concerns remain. Please see details as follows.
>
> >our work identifies a key gap between attention-based Transformers (which store K and V for all seen samples) and suitable methods for meta-continual learning (MCL).
>
> I agree the authors have clearly explained these issues for MCL. But I would argue that the inefficiencies associated with the KV-cache and memory utilization are more of an intrinsic issue related to the Transformer architecture itself—specifically, the softmax attention mechanism. This challenge is not isolated to the MCL setup; it is a common issue across many applications of Transformers. In fact, attention-free models like Mamba have been proposed to address these challenges more generally. I would appreciate your thoughts on this perspective.
>
>
> >On technical aspect, we introduce the attention-free Mamba model tailored to the MCL formulation and propose specific techniques to ensure its effectiveness. This represents a novel contribution.
>
> In addition to the above, more importantly though, it remains unclear how this represents a significant departure from merely applying Mamba to the MCL context.
>
> The authors mentioned *"introduce the attention-free Mamba model tailored to the MCL formulation and propose specific techniques to ensure its effectiveness"*, could the authors perhaps elaborate more on the specific changes (tailoring) that were necessary for Mamba to be effectively applied to MCL (besides the regularization technique which we can discuss below)? I think this would be particularly helpful in illustrating these points.
>
> >Unlike (Lee et al., 2024), where attention-free methods fail to match the performance of Transformers, our work demonstrates the successful development of an attention-free model for MCL...
> >
> >... such as generalization to diverse meta-test tasks and robustness to noisy scenarios. These are novel perspectives and distinguish our approach from
>
> Off-course, I am aware one technical change being made is the regularization proposed in this paper.
>
> While the regularization technique does contribute to stabilizing the meta-training process, I am concerned that the observed results (regarding better generalization of Mamba) may largely stem from the use of the Mamba model itself rather than this algorithmic development. Specifically, my reasoning includes:
> - This technique appears to have very limited effectiveness on the attention-free model beyond Mamba, which still resulted in  significant meta-overfitting.
> - The authors mentioned that generalization performance is relatively insensitive to the strengths of the regularization (fig 9,3). This raises questions about the role of the regularization technique in achieving the reported improvements for Mamba.
>
> As a results, would you agree that the empirical performance improvements observed in comparison to Transformers — specifically in memory efficiency and generalization to longer input sequences — are more likely inherent properties of the Mamba model itself, which were already highlighted in the original Mamba paper?
>
>
> >Due to the complexity of the MCL task, the proposed regularization technique plays a crucial role in stabilizing and improving the training process for all models.
>
> More on this regularization technique. I agree that it stabilizes meta-training, but I am still not sure if I understood the rationale behind applying this technique to all models, as this would prevent us from seeing how different models behave intrinsically.
>
> Initially, I thought meta-training was impossible without this technique, but it seems that Lee et.al 2024 managed to produce meaningful results without this technique. Although the meta-training losses, in the new Figures, showed more oscillation, they still showed a clear decreasing trend indicating convergence. I do not require additional experiments on this point, but I would appreciate your insights.
>
> >... We aim to address these challenges from the broader perspective of extending MCL to more realistic and practical scenarios...
> > In this study, our goal is not to resolve these challenges using a single specific model, such as Mamba.
>
> > ...  we focus on the studies of attention-free MCL models, which sets our work apart from (Lee et al., 2024) and provides a novel direction for MCL research.
>
> I have some reservations about the work provides a novel direction for MCL in the use of attention-free methods. As previously mentioned, the improvements attributed to the empirical improvements over transformers seem to be closely tied to the specific characteristics of the Mamba model. The significant meta-overfitting observed in other attention-free methods suggests that the broader applicability of these models for MCL may be limited by the specific configurations of your meta-training setup.

---

> ### Author Response · Authors · 2024-11-29
> **Thank you for your comments (Round 2 Part 1/3)**
>
> We sincerely appreciate the time and effort you have dedicated to reviewing our manuscript. However, we noticed that some misunderstandings may have influenced certain aspects of the review. We further clarify all the questions and try to address all concerns more straightforwardly.
>
> > **Q1. ...inefficiencies associated with the KV-cache and memory utilization are more of an intrinsic issue related to the Transformer architecture itself...This challenge is not isolated to the MCL setup; it is a common issue across many applications of Transformers. In fact, attention-free models like Mamba have been proposed to address these challenges more generally...**
>
> - For the general purpose of the sequence model, standard Transformers have an efficiency issue, but they are still practical. However, for MCL, we want to highlight that – although Transformers can be applied to the sequence and produce good-looking performances, they actually violate the intrinsic requirements in CL. Transformers maintain the representations (i.e., K and V) of all seen samples, which contradicts the objectives of CL.
> - The most critical issue of Transformers in MCL is its misalignment with the motivation, not only the efficiency issue. We thus do not agree that `“... isolated to the MCL setup; it is a common issue ...”`.
> - For the above motivation, we focus on attention-free methods, which have the formulation aligning with the requirements of MCL. However, all the tested attention-free methods in previous work (Lee et al., 2024) cannot perform well. We thus investigate `“whether the attention-free Mamba can perform well for MCL”`, as discussed in the introduction. This is one important perspective we want to highlight, which is essential for this MCL research area and not obvious.
> - Directly applying Mamba on MCL is not trivial due to the difference in model structure between Mamba and Transformers and the difference between MCL and other standard sequence modeling tasks. Specific techniques are contributed. Moreover, we expand the standard MCL formulation in (Lee et al., 2024) to more realistic MCL scenarios and reflect them into experimental settings, such as the generalization analysis of novel meta-testing cases, robustness analysis of noisy cases, and cross-domain scenarios. Although the success of Mamba on language-based tasks has been observed, it is not trivial and direct to achieve success on MCL. In the general experiments, we observed Mamba can align the performance of Transformers and even outperform them in some scenarios (with explainable reason aligning Mamba’s success on some specific language-based tasks), which is specific to our work.
>
>
>
> ---
>
> > **Q2. ...unclear how this represents a significant departure from merely applying Mamba to the MCL context...could the authors perhaps elaborate more on the specific changes (tailoring) that were necessary for Mamba to be effectively applied to MCL?**
>
> - Firstly, We want to highlight that it is not a trivial task to adapt Mamba to MCL (even more challenging than applying Transformer to MCL), especially with the background in (Lee et al., 2024) that none of the attention-free methods (which better aligns the requirement of MCL) can work well. We are the first to make Mamba or arbitrary attention-free methods work well on MCL.
> - Secondly, we introduce the association regularization for Mamba by bridging the selective operation of Mamba with attention operation in Transformer. The attention regularization for Transformer can be straightforward. The regularization for Mamba is not obvious. We appreciate that the reviewer has recognized the contribution and novelty.
> - Additionally, we investegated how the hidden state capacity in SSM/Mamba influences the resutls in MCL, considering that Mamba and SSM selectivgely compress the context information of seen samples in the data stream. We also explore the possibility of integration of Mixture of Experts (MoE) with Mamba to enhance learning and mixing multiple learning components.

---

### Author Response · Authors · 2024-11-23
**Response by authors**

We sincerely thank all reviewers for their time and effort in reviewing our manuscript.
We have considered each comment and have responded to each reviewer individually. An updated version of the manuscript has been uploaded accordingly.

---

### Meta-Review · Area_Chair_d8rp · 2024-12-22

**Metareview:**

The paper builds on the meta continual learning (MCL) framework proposed by Lee et al., 2024, by introducing Mamba as the sequential model to replace transformers or attention-free variants. This substitution aims to reduce computational costs while maintaining competitive performance. Furthermore, the authors propose a selective regularization technique during meta-training to strengthen the connection between query tokens and previously correlated input tokens. Experimental results indicate that Mamba achieves comparable generalization and robustness to Transformer variants in MCL, while requiring less memory during inference.

The paper’s strengths include its focus on an important research question. However, the work suffers from significant weaknesses, including limited technical novelty and marginal contributions to advancing the field.

Given these limitations, I recommend rejecting this submission.

**Additional Comments On Reviewer Discussion:**

During the discussion period, Reviewers Zeh1 and BA8y actively engaged with the authors, while Reviewer bJ6d acknowledged reading the responses.

After carefully reviewing all the concerns raised by the reviewers, I found that the authors failed to adequately address several critical issues, including key points highlighted by Reviewer bJ6d.

A central issue raised by all reviewers pertains to the lack of technical novelty in the proposed method. Specifically, the reviewers were unsurprised by the effectiveness of replacing Transformers in (Lee et al., 2023) with Mamba, given (1) Mamba’s established effectiveness in sequence modeling and (2) the recasting of MCL as sequence modeling in (Lee et al., 2023). Unfortunately, the authors did not provide a convincing response to this concern.

One of the primary concerns from Reviewer Zeh1 relates to the confounding effect of the proposed regularization. Notably, Reviewer Zeh1 made an effort to help the authors articulate their key technical novelty, suggesting that it should go beyond the straightforward application of Mamba to (Lee et al., 2023). While the authors argued that this application is non-trivial, their claim appears to rely heavily on the proposed regularization. However, the authors themselves acknowledged that "such regularization is also effective for other models for convergence, as observed in (Lee)," which further undermines its originality.

These unresolved concerns significantly restrict the scope and potential impact of this work, limiting its appeal to the broader community. As such, I believe it does not meet the high standards required for acceptance at the prestigious ICLR conference.

---

### Decision · Program_Chairs · 2025-01-22

Reject